

# Speeding up large wind farms layout optimization using gradients, parallelization, and a heuristic algorithm for the initial layout

Rafael Valotta Rodrigues[1], Mads Mølgaard Pedersen[1], Jens Peter Schøler[1], Julian Quick[1], and Pierre-Elouan Réthoré[1]

[1]Technical University of Denmark, Frederiksborgvej 399, 4000 Roskilde, Denmark

**Correspondence:** Rafael Valotta Rodrigues (ravaro@dtu.dk)

**Abstract.**

As the use of wind energy expands worldwide, the wind energy industry is considering building larger clusters of turbines. Existing computational methods to design and optimize the layout of wind farms are well suited for medium-sized plants; however, these approaches need to be improved to ensure efficient scaling to large wind farms. This work investigates strate-
gies for covering this gap, focusing on Gradient-Based (GB) approaches. We investigated the main bottlenecks of the problem, including the computational time per iteration, multi-start for GB optimization, and the number of iterations to achieve convergence. The open-source tools PyWake and TOPFARM were used to carry out the numerical experiments. The results show Algorithmic Differentiation (AD) as an effective strategy for reducing the time per iteration. The speedup reached by AD scales linearly with the number of wind turbines, reaching 75 times for a wind farm with 500 wind turbines. However, memory
requirements may make AD unfeasible in personal computers or for larger farms. Moreover, flow case parallelization was found to reduce the time per iteration, but the speedup remains roughly constant with the number of wind turbines. Therefore, top-level parallelization of each multi-start was found to be a more efficient approach for GB optimization. The handling of spacing constraints was found to dominate the iteration time for large wind farms. In this study, we ran the optimizations without spacing constraints and observed that all wind turbines were separated by at least 1.4D. The number of iterations until
convergence was found to scale linearly with the number of wind turbines by a factor of 2.3, but further investigation is necessary for generalizations. Furthermore, we have found that initializing the layouts using a heuristic approach called Smart-Start (SMAST) significantly reduced the number of multi-starts during GB optimization. Running only one optimization for a wind farm with 279 turbines initialized with SMAST resulted in a higher final AEP than 5,000 optimizations initialized with random layouts. Finally, estimates for the total time reduction were made assuming the trends found in this work for the time per itera-
tion, number of iterations, and number of multi-starts holds for larger wind farms. One optimization of a wind farm with 500 wind turbines combining SMAST, AD, flow case parallelization, and without spacing constraints takes 15.6h, whereas 5,000 optimizations with random initial layouts, finite-differences, spacing constraints, and top-level parallelization are expected to take around 300 years.





# 1 Introduction: Wind Farm Layout Optimization

The use of wind energy worldwide increases year by year. The global cumulative wind power capacity reached 837GW by the end of 2021, with a prediction of around 3,200GW by 2030 (GWEC, 2022). This growth opens the path for larger wind farms. Existing literature shows a gap in approaches and strategies to efficiently perform Wind Farm Layout Optimization (WFLO) with hundreds of wind turbines. A proper framework to address the problem could enable faster evaluation of thousands of different configurations, allowing trade-off sensitivity analysis and design insights in a more extensive and faster way.

Since the first work on WFLO by Mosetti et al. (1994) using a Gradient-Free (GF) approach, the literature on the topic massively evolved around GF. GF-based approaches on the topic include metaheuristic methods such as Genetic Algorithm (GA) (González et al., 2018; Wang et al., 2015; Parada et al., 2017), Particle Swarm Optimization (PSO) (Hou et al., 2016; Pillai et al., 2017; Veeramachaneni et al., 2012; Wan et al., 2012; Pookpunt and Ongsakul, 2016), Random-Search (RS) (Feng and Shen, 2017b, a), and many others. GF methods explore the entire design space and may find the global optimum at some point, but processing time explodes with the number of design variables. Therefore, GF methods tend not to scale well for problems with many design variables (Martins and Ning, 2021; Ning et al., 2019) and are more suitable for smaller problems (Wright et al., 1999). Gradient-Based Wind Farm Layout Optimization (GBWFLO) has been explored more since a few years ago. Research in the field has been evolving, including analytical computation of the gradients (Guirguis et al., 2016, 2017; Stanley et al., 2019), a quasi-Newton limited-memory optimizer called Limited-memory Broyden-Fletcher-Goldfarb-Shanno (L-BFGS) that estimates the inverse of the Hessian Matrix using a generalized secant method (van Dijk et al., 2017; Croonenbroeck and Hennecke, 2021), another limited-memory optimizer called SNOPT (Sparse Nonlinear Optimizer) that explores the sparsity of the Jacobian matrix (Tingey and Ning, 2017), SNOPT with Finite Differences (FD) (Fleming et al., 2016), SNOPT with analytical gradients (Gebraad et al., 2017), and Adjoints (King et al., 2017; Allen et al., 2020). Mittal et al. (2016) developed a hybrid GF (GA) and GB (fmincon) algorithm.

Literature has few articles comparing GF and Gradient-Based (GB) in a systematic and standardized way (e.g., same configurations). Brogna et al. (2020) performed WFLO in complex terrain with 25 wind turbines, comparing six GF with two GB methods that use Global Search and Multi-start from MATLAB optimization toolboxes. They reported GF (RS, Pattern Search, and Local Search) outperforming the two GB approaches analyzed on both computational costs and optimization results. Croonenbroeck and Hennecke (2021) performed layout optimization to maximize profit and efficiency, which is the ratio between the Annual Energy Production (AEP) and theoretical maximum AEP. They compared a GB (L-BFGS-B) against GF algorithms, including modified versions of GA, PSO, Simulated Annealing (SA), and RS. They found L-BFGS-B to perform the fastest among all the options but not with the best results in AEP on a spectrum of six runs. Guirguis et al. (2016) compared GB with analytical derivatives and GA. Additionally, the study compared two GB Interior Point Method (IPM) approaches where one used FD to compute the gradients and the other used exact analytical gradients. They found the computational costs of the GB FD approach to be around 20 times higher than the GB with analytical gradients. Additionally, the GB with analytical gradients resulted in 0.36% higher wind farm efficiency. Even though the literature is not in total agreement on which approach is the best, GB methods are worthy of further development, especially for large WFLO with many design variables. In order



to make GBWFLO more efficient and applicable for large wind farms, the associated computational cost and time need to be properly addressed. The next section will break down the computational cost into different components. In the following
sections, approaches to reduce the computational cost are proposed.

## 2 Total Computational Time for Gradient-Based (GB) Optimization

Equation (1) shows the total computational time to perform GB optimization. To accomplish faster large GBWFLO, one needs to tackle the bottlenecks of the problem, which are the variables in Equation (1).

$$t_{total} = (t_{iter} \cdot n_{iter} + t_{init}) \cdot \frac{n_{multistarts}}{n_{cpu}} \tag{1}$$

where $t_{total}$ is the total computational time for the GBWFLO, $t_{iter}$ is the time per iteration, $n_{iter}$ is the number of iterations until convergence, $t_{init}$ is the time to initialize the problem, including time to generate the initial layout, e.g. via Smart-Start (SMAST) from section 3.4.1, $n_{multistarts}$ is the number of initial starts to avoid getting stuck in local minima, as visually demonstrated in (Thomas et al., 2022b), and finally, $n_{cpu}$ is the number of CPU cores available for parallelization of the $n_{multistarts}$ independent optimizations with different initial layouts.

### 2.1 Time per Iteration

In GB optimization, each iteration typically consists of computing the gradients of the objective function and constraints with respect to all design variables, followed by a line-search that comprises one or more function evaluations of the objective and constraints. In this section, we analyze different approaches to reduce the iteration time of GBWFLO.

### 2.1.1 Gradients Computations: Analytical vs Finite-Differences

Computing gradients can be done with different methods. In Algorithmic Differentiation (AD), all the lines of code are differentiated. These lines are usually composed of simple mathematical operations. AD performs differentiation with respect to each relevant variable at each line of code by applying the chain rule, and then sums up all the contributions. The FD method computes the derivatives using a Taylor series expansion, as shown in Equation C1 (Appendix C). FD computes the Jacobian matrix by looping through all the dimensions to compute the function values, perturbing with a determined step size, and com-
puting the differences in the function. The value of the step size dictates the truncation error. Smaller step sizes reduce the error but increase the amount of numerical noise. The Complex-Step (CS) method also relies on a Taylor expansion to compute the derivatives. However, the step is represented by an imaginary term in the complex plane (Equation C3, Appendix C). The CS method typically doubles computational time, as there are two times more bits in each value. The advantage of the CS is that the only source of error is the truncation error since there is no associated subtraction cancellation error. Adopting smaller step
sizes can reduce truncation errors.





### 2.1.2 Parallelization

One of the most common objective functions in WFLO is the AEP, which is computed by summing up the contributions of the various combinations of wind direction sectors and wind speeds (referred to as flow cases). The number of flow cases during each iteration is a function of the discretization of the wind resource. To avoid numerical discrepancies, it is necessary to

discretize finely the bins of wind directions and wind speeds. The number of flow cases can sum up to 8280 if wind directions and wind speed bins of 1°(0°to 360°) and 1m/s (3 to 25m/s) are considered, for instance. The contribution of each flow case to the AEP is multiplied by the frequency of occurrence of that combination, and all of them are summed up sequentially in one CPU to calculate the total AEP. As the flow cases are independent, parallelization could speed up iterations as the calculation of AEP contributions of each flow case can be done simultaneously on several CPUs rather than sequentially on one CPU.

Throughout the text, this is referred to as flow case parallelization.

### 2.1.3 Constraints

Handling constraints in GBWFLO can be done with penalty functions, Sequential Quadratic Optimization (SQP), and IPM (Martins and Ning, 2021). Constraints in GBWFLO usually include physical boundaries and minimal spacing between turbines. Looking at the literature on WFLO, the spacing constraint between turbines varies. Many studies consider 5 wind turbine rotor

diameters (D) (Gao et al., 2015; Wang et al., 2015; Parada et al., 2017), but others consider 4D (Hou et al., 2016; Rodrigues et al., 2015), 3D (Mittal et al., 2017; Abdulrahman and Wood, 2017), 2D (Stanley and Ning, 2019; Padrón et al., 2019; Kirchner-Bossi and Porté-Agel, 2018; Gebraad et al., 2017; Fleming et al., 2016), and a few works consider values beyond 5D (Rodrigues et al., 2016). When there are too many constraints in optimization, for instance, large WFLO with wind turbine pair-spacing constraints, combining all the constraints into a single constraint is also possible (Martins and Ning, 2021).

### 2.2 Number of Initial Starts

Better initial guesses for the layout can potentially avoid the worst local optima. In previous literature, GF has been found in some studies to provide better results than multi-start GB. However, just a few starts were applied for the GB, and the layouts were randomly initialized. It is unclear, for instance, if the work done by Croonenbroeck and Hennecke (2021) to optimize a wind farm with 20 wind turbines could have found L-BFGS outperforming (in AEP) the GF methods if more multi-start runs were

applied (six runs in this study). Examples of previous studies on multi-start for GBWFLO considered random multi-start (Brogna et al., 2020; Yang and Deng, 2023; Thomas et al., 2022a; Baker et al., 2019) and Latin-Hypercube sampling (Guirguis et al., 2016), while an example of a multi-start GF (RS) also using randomly produced initial guesses is given in Feng and Shen (2017a). A heuristic approach developed by Perez et al. Pérez et al. (2013) assumed a turbine widespread throughout the wind farm area as a strategy to avoid the wake effects and produce better initial guesses. They generated random regular rectangles,

applied a rectangular transformation to extend the points to the wind farm boundaries, and used triangulations to maximize the sum of the areas of the triangles. Still, they had to generate a set of random candidate solutions to initiate the approach. One could enhance heuristic approaches for the initial layout with physics information (e.g., wake effects), avoiding random



guesses for the layout. These examples show that there is room for improving multi-start GB, for instance, if layouts were initialized using some threshold or method based on physics, rather than purely random. Smart-Start (SMAST) is a heuristic approach based on physics, where wind turbine wake effects guide the decisions to place wind turbines for the initial layout sequentially. SMAST is now available in PyWake (Pedersen et al., 2022), and here we show to which extent the method can improve multi-start for large GBWFLO. That is potentially a way of improving multi-start for GB optimization of wind farms, one of the objectives of this present study.

### 2.3 Objective and Scope of the Work

This work aims to speed up layout optimization for large wind farms. The strategy is to tackle the bottlenecks in terms of computational expenses. Specifically, on optimization iteration time, the objective is to show how different GB approaches and parallelization of the flow cases scale with the Number of Wind Turbines ($n_{wt}$). Moreover, we explore a heuristic approach to produce better initial layout guesses and improve multi-start, which is necessary for GB methods.

### 2.4 Contributions to the Existing Literature

This work intends to complement and add the following new insights to the existing literature:
- Evaluate how parallelization scales with $n_{wt}$ during large GBWFLO.
- Evaluate how different techniques to compute gradients scale with $n_{wt}$ when performing GBWFLO
- Evaluate how $n_{iter}$ scale with $n_{wt}$ when performing GBWFLO.
- Demonstrate how $n_{multistarts}$ scales with $n_{wt}$ and how to reduce this number using a heuristic approach.

## 3 Methods

The AEP computations and the optimizations were performed in PyWake (Pedersen et al., 2023) and TOPFARM (Réthoré et al., 2014), which are open-source tools developed by the Technical University of Denmark. Examples of previous works using PyWake and TOPFARM can be found in the literature (Rodrigues et al., 2022; Ciavarra et al., 2022; Criado Risco et al., 2023; Quick et al., 2022; Pedersen and Larsen, 2020; Nyborg et al., 2023; Fischereit et al., 2021; Pérez-Rúa and Cutululis, 2022). Section 3.1 provides all the relevant details about the optimization formulation. Section 3.2 shows the case studies considered in this work. Section 3.3 describes the methods and assumptions to evaluate iteration time, whereas Section 3.4 provides an overview of the heuristic method to produce efficient initial layouts and improve multi-start GBWFLO.

### 3.1 Optimization: Problem Formulation

For the AEP computations, an implementation of a Gaussian wake model (Bastankhah and Porté-Agel, 2014) referred to as Bastankhah Gaussian (BG) in Table 1 was considered. The optimization algorithm used in this work is the Sequential Least-Squares Quadratic Programming (SLSQP), which relies on quasi-Newton methods (Powell, 1964; Liu and Nocedal, 1989) and is suitable for constrained GB optimization problems (Wu et al., 2020; Perez et al., 2012; Virtanen et al., 2020). The



optimization in this work follows the formulation in Equation (2), where the objective function is the AEP, and the design variables are the layout coordinates $x$ and $y$ of the turbines. Moreover, farm boundary constraints are applied to restrict the area upon which the turbines can move. TOPFARM handles the design variables and constraints corresponding to the problem formulation in Equation (2).

$$\max_{x,y} \quad AEP(x,y) \approx 8760 \sum_{d=1}^{N_\theta} \sum_{u=1}^{N_u} P_{d,u}(x,y) \cdot \rho_{d,u} \tag{2}$$

$$\text{s.t.} \quad C_{kj} \geq 0 \forall k,j$$

where $C$ is a matrix of wind farm boundary constraint, $d$ denotes wind directions, $u$ refers to wind speeds, with $N_\theta$ and $N_u$ standing for the number of wind directions and wind speeds; $P_{d,u}$ represents the power output of the wind farm given by the wind turbine coordinate vectors $x$ and $y$, for wind direction $d$ and inflow wind speed $u$; lastly, $\rho_{d,u}$ is the frequency of wind direction $d$ and inflow wind speed $u$.

When the wind farm is circular, the boundary constraint, $C$, is a $1 \times n_{wt}$ matrix, defined in Equation (3):

$$C_{k,1} = R_{wf} - \sqrt{x_k^2 + y_k^2}, \tag{3}$$

where $k$ is an integer denoting the turbine number, and $R_{wf}$ is the Wind Farm Radius.

When the wind farm span is a parallelogram, the boundary constraints, $C$, are defined as a $4 \times n_{wt}$ matrix,

$$C_{k,1} = \left[ \left( \frac{x^{UR} - x^{LR}}{y_{\max} - y_{\min}} \right) (y_k - y_{\min}) + x^{LR} \right] - x_k \tag{4}$$

$$C_{k,2} = x_k - \left[ \left( \frac{x^{UL} - x^{LL}}{y_{\max} - y_{\min}} \right) (y_k - y_{\min}) + x^{LL} \right] \tag{5}$$

$$C_{k,3} = y_{\max} - y_k \tag{6}$$

$$C_{k,4} = y_k - y_{\min} \tag{7}$$

where $x^{UL}$, $x^{UR}$, $x^{LL}$, and $x^{LR}$ are the upper left, upper right, lower left, and lower right coordinates that respectively define the parallelogram boundaries.

As pointed out in section 2.1.3, adopted values of spacing constraint between wind turbines in a wind farm vary in the literature. A spacing constraint value of 2D was adopted in this work, aiming to not over-constrain the problem. The spacing constraint, though, was applied only to the initial layout (Equation 8) generated by the heuristic method described in section 3.4.1. For the remaining optimization, spacing constraints were disregarded and the formulation of Equation (2) was adopted. This setup for the spacing constraint is adopted because the cost of handling spacing constraints for each turbine pair does not scale well with $n_{wt}$. Additional discussion around the spacing constraint consideration in this work is provided in section 4.1.3, where we provide a plot showing the influence of spacing constraints on speeding up GBWFLO across scales. Moreover, we show and discuss in the Discussion section that the minimum spacing in our final results is at least 1.4D.





$$\max_{x,y} \quad AEP(x,y) \approx 8760 \sum_{d=1}^{N_\theta} \sum_{u=1}^{N_u} P_{d,u}(x,y) \cdot \rho_{d,u}$$

$$\text{s.t.} \quad C_{kj} \forall k, j \geq 0$$

$$\sqrt{(x_i - x_j)^2 + (y_i - y_j)^2} \geq 2D$$

(8)

where the sub-indices $i$ and $j$ denote spatial locations.

## 3.2 Study-Cases

Two study cases have been considered in this work, which are summarized in Table 1. The power and $C_T$ curves are provided in Figures 1a and 1b, respectively. The simulations to investigate the time per iteration were performed with a realistic setup, the Horns Rev I. A Weibull distribution was fitted to the local wind resource (Figure 1c). We further extended the analysis of Horns Rev I to assess how the approaches tested in this work scale with the number of wind turbines $n_{wt}$ (Figure 2).

The results for the number of iterations and number of multi-starts are based on more than 50,000 GBWFLOs, and these were performed with a faster setup, which uses the wind turbine, site, and wake model definitions from the IEA Wind Task 37 case study 1 (IEA Wind Task 37, 2018). Extra cases were designed to scale the analysis (Figure 3). In this simplified setup, only the rated wind speed (9.8m/s) is simulated, and all wind turbines operate with constant $C_T \approx 0.964$. The wake model for these simulations is the Bastankhah Gaussian (BG) with a constant $C_T \approx 0.964$ (called Simple Bastankhah Gaussian - SBG at Table 1). Another reason for choosing the IEA 37 site is related to the circular boundaries, as rectangular boundaries use four times more boundary constraints. Furthermore, Figure 1d shows the frequency of occurrence of wind directions for the IEA 37 site. Note that, in the current study, we simulate 360 wind directions (as shown in Table 1) while only 16 wind directions were considered in the original IEA Wind Task 37 case study. Our results are, therefore, not directly comparable to the AEP results of the benchmark in Baker's work (Baker et al., 2019). Even though the absolute values (such as AEP) are not consistent with the original IEA 37, we expect our results on $n_{iter}$ and multi-starts, as well as the relative comparisons presented in this study, to be valid for more realistic wind farm setups as well.

**Table 1.** Cases considered for the optimization and setup/models used in PyWake/TOPFARM.

| Site | $n_{wt}$ | Wind Turbine | Wake Model | Superposition | WS Bins | WD Bins |
|------|----------|--------------|------------|---------------|---------|---------|
| IEA37 | 16, 36, 64, 130, 279, 566 | IEA37 3.35MW | SBG | Squared Sum | 1 | 360 |
| Horns Rev I | 100, 200, 300, 400, 500 | V80 2MW | BG | Squared Sum | 23 | 360 |

## 3.3 Time per Iteration: Horns Rev I

This section provides the methods implemented to accelerate iteration time during the optimization described in Section 3.1. These simulations use a realistic setup (Horns Rev I), as described in section 3.2 and summarized in Table 1.



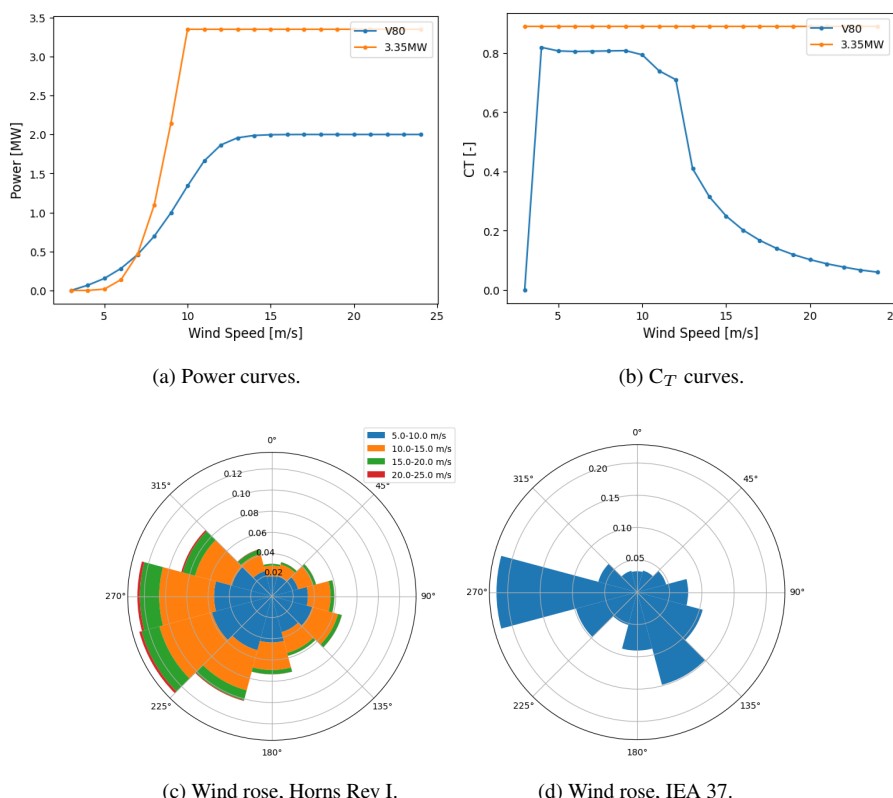

(a) Power curves.

(b) $C_T$ curves.

(c) Wind rose, Horns Rev I.

(d) Wind rose, IEA 37.

**Figure 1.** Power curves, CT curves, and wind rose distribution for the sites considered in this study.

### 3.3.1 Gradients Computation

We tested different techniques to compute the gradients and evaluate how they scale with $n_{wt}$, including AD, FD, and CS. The theoretical background of these different gradient methods can be found in section 2.1.1, Appendix C, or in the literature

200 (Martins and Ning, 2021). In this work, we have adapted PyWake to use "Autograd" (Maclaurin et al., 2015) and perform AD to automatically calculate the gradients of the output (AEP) with respect to the inputs (layout coordinates x and y).

### 3.3.2 Parallelization of the Flow Cases

In this work parallelization is studied, these studies are performed on a computational cluster. Single node operation was utilized with each node being composed of 2x AMD EPYC 7351 16 core CPUs, @2.9GHz, with 128GB of RAM. All the

205 CPUs in each node (32 CPUs per node) are set up to run one simulation. PyWake parallelizes the flow cases, computing chunks of wind directions and wind speeds throughout the several CPUs within a node.



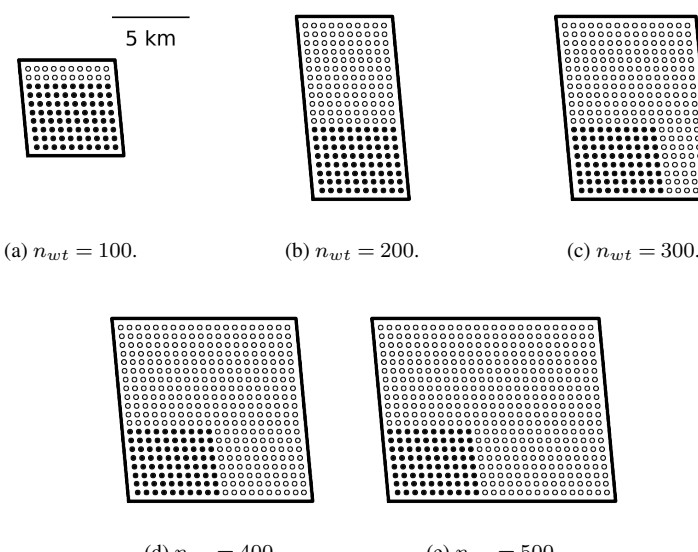

**Figure 2.** Considered variations of the scaled Horns Rev I site.

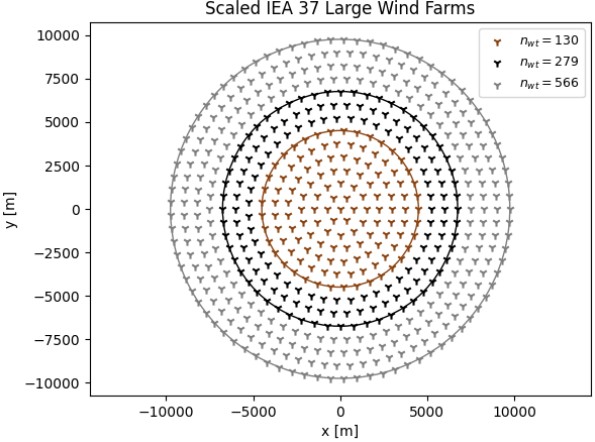

**Figure 3.** Scaled IEA 37 layouts with $R_{wf}$ of 4500m, 6750m, and 9750m with $n_{wt}$ of respectively 130, 279, and 566.

## 3.4 Number of Initial Starts: IEA 37

As the total computational time for GB is a function of $n_{multistarts}$, this section provides methods for investigating that bottleneck. This work explores a heuristic (SMAST) to efficiently generate initial layouts for WFLO. The objective of the method is to speed up WFLO by reducing the number of multi-starts necessary to achieve optimized solutions. Section 3.4.1 details SMAST and section 3.4.2 presents a methodology to improve multi-start GBWFLO based on a comparison between SMAST and a random set of simulations.



### 3.4.1 The Smart-Start (SMAST) Algorithm: A Heuristics Method

The objective of the SMAST is to provide a better initial layout for multi-start GBWFLO. The process is described in Algorithm
1. First, SMAST defines an array $\mathcal{L}$ with all the potential positions for wind turbines, in this case a regular grid of points
covering the domain. Next, SMAST removes positions from $\mathcal{L}$ not satisfying constraints, i.e., farm boundary constraints.
SMAST then computes the AEP at all the remaining points in $\mathcal{L}$, considering the wakes from the turbines previously added.
Next, SMAST randomly selects a point $p$ among the points associated with the highest AEP, $\updownarrow_{best}$, and places the next turbine
at $p$. Finally, SMAST removes $p$ and all the points that violate the spacing constraint of the newly added wind turbine. This
process is repeated until all wind turbines have been placed. As described, SMAST ignores the wake effects of the turbine to
be added, i.e., the power reduction of the already added turbines due to wake effects from the new turbine is ignored. This
simplification is, however, necessary to make the method feasible. SMAST has a parameter to define the desired degree of
randomness ($random_{pct}$) when selecting the point $p$. If SMAST is run without randomness ($random_{pct} = 0$), the algorithm
places the turbine at the point with the highest AEP. In case multiple points provide the highest AEP, e.g. in the first iteration
assuming a uniform site, the algorithm randomly selects one of these points. This means that even for $random_{pct} = 0$, the
algorithm is able to provide different layouts. SMAST with some randomness ($random_{pct} > 0$) takes more possibilities for
$_{best}$ points and picks one of them randomly. The higher is the $random_{pct}$, the more $_{best}$ points are considered. If SMAST
is entirely random ($random_{pct} = 100$), SMAST bypasses the AEP calculation and quickly generates a random layout that
satisfies the boundary and spacing constraints. Another parameter that influences the AEP provided by the SMAST is the
resolution of the grid defined in the Algorithm 1. Figures D1 and D2 show examples of SMAST AEP flow maps of the
potential positions $\mathcal{L}$ with grids with resolutions of 3R and 6R (i.e. the distance between points in $\mathcal{L}$), respectively. Figures D3
and D4 show how the AEP and the computational time vary according to the grid size. The finer the SMAST grid, the higher
the AEP (except for $n_{wt} = 16$ where it stabilizes after 4R) but also the higher the computational time of SMAST. Memory can
also be a problem in running SMAST if the grid is too refined.

---

**Algorithm 1** Smart-Start (SMAST) Algorithm

---

1: $\mathcal{L} \leftarrow$ {Potential positions for wind turbines}

2: $\mathcal{L} \leftarrow \mathcal{L}$ - OOB$_{\mathcal{L}}$ {**OOB: Out of Boundary positions**}

3: $\boldsymbol{P} \leftarrow$ {initialize empty turbine position vectors}

4: **for** $i = 1$ *to* $n_{wt}$ **do**

5: $\quad Z_{\mathcal{L}} \leftarrow \texttt{AEP}\,(\mathcal{L}, P)$

6: $\quad min_{AEP} = random_{pct}{}^{th}$ *percentile of* $Z_{\mathcal{L}}$

7: $\quad l_{best} = l \in \mathcal{L},$ *where* $Z(l) \geq minAEP$

8: $\quad p \in_R l_{best},$ *where* $\in_R$ *selects a random element*

9: $\quad \mathrm{P} = \mathrm{P} \cup \mathrm{p}$

10: $\quad$ *drop elements of* $\mathcal{L}$ *where* $dist(\mathcal{L}, p) < 2D$

11: **end for**

---





### 235   3.4.2   Random Multi-Start versus SMAST

Aiming to showcase the capabilities of SMAST to improve multi-start for GBWFLO, we consider sets of random multi-start simulations with three different IEA 37 case studies: 16, 64, and 279 wind turbines. Those sets are the baseline for the comparisons. The methodology in this work consisted of running 10,000 random simulations for each case (i.e., randomly generated initial layouts), splitting the results into m chunks, and computing the maximum AEP of each chunk. Finally, the

mean and confidence intervals of the m maximum values are computed. Figure 4 shows how the normalized optimized AEP varies with the number of random initial starts for the three cases. The bandwidth in the plots represents the standard deviation within a 99% confidence interval of *th* mean. To better clarify the methodology, let us look at the 16 wind turbines case at 1,000 initial starts. The procedure consists in splitting the 10,000 simulations into 10 chunks of 1,000 simulations, computing the maximum AEP of each of the 10 chunks, and computing the mean and 99% confidence interval of these maximum values

with Equation (9). The results from Figure 4 are going to be used in section 4.3, where we showcase how SMAST improves GBWFLO by achieving the same final optimized results as the random approach (to generate the initial layout) but with a reduced number of multi-starts. Furthermore, what is noticeable in Figure 4 is that 99.9% of the maximum AEP is obtained around 500 starts for $n_{wt} = 16$, around 2,500 starts for $n_{wt} = 64$, and around 4,000 starts for $n_{wt} = 279$. For small problems, the random approach seems to work as the maximum AEP converges after a relatively low number of starts. Other methods

are necessary for larger problems, as the example with $n_{wt} = 279$ shows that no full convergence is achieved even after 5,000 starts.

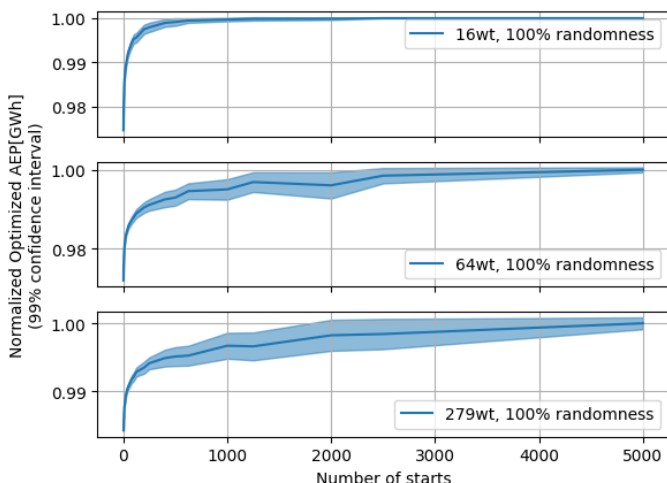

**Figure 4.** Simulations with initial layout randomly generated for three cases of the IEA 37: 16, 64, and 279 wind turbines. These are the baseline cases for comparison with SMAST.

$$AEP_{ub,lb} = AEP_{mean} \pm 2.576 \cdot \frac{\sigma}{\sqrt{m}} \tag{9}$$





where $AEP_{ub}$ and $AEP_{lb}$ are the upper and lower bounds of the AEP within a 99% confidence interval, m is the number of chunks, $AEP_{mean}$ is the mean of the maximum AEP value of each chunk, and $\sigma$ is the standard deviation of the maximum

AEP values.

## 4 Results and Discussion

In this section, we present the results of our study on speeding up GBWFLO by exploring each of the variables in Equation 1. Section 4.1 shows how different gradient computation methods (section 4.1.1) and parallelization (section 4.1.2) impact $t_{iter}$. Additionally, the influence of spacing constraints on $t_{iter}$ is shown in section 4.1.3. Section 4.2 how $n_{iter}$ scales with $n_{wt}$.

Section 4.3 shows how SMAST can improve $n_{multistarts}$ for GBWFLO and finally section 4.4 addresses the impact of these findings on the total optimization time.

### 4.1 Time per Iteration

This section explores $t_{iter}$ during GBWFLO, as well as strategies to speed up $t_{iter}$ by different gradient computation methods and parallelization of the flow cases. These results are based on the more realistic Horns Rev 1 setup.

### 4.1.1 Impact of Gradient Computation Method

Figure 5a shows the time per iteration for AD, FD, and CS, whereas Figure 5b shows the speedup when comparing these gradient methods to FD. According to Figures 5a and 5b, AD computes gradients faster than FD and CS, especially as $n_{wt}$ increases. AD is around 20 times faster than FD for 100 wind turbines and the speedup increases roughly linearly with $n_{wt}$. This is expected and confirms that FD is only feasible for optimizing small wind farms. Figure 5c shows the *Memory Usage*

for each method, revealing a trade-off between speed and memory. As $n_{wt}$ increases, AD consumes more memory than FD and CS. AD memory usage is 85Gb for 500 turbines, which is usually beyond a regular computer's configuration. This value (85Gb) is more than four times higher than CS and around 5 to 6 times higher than FD. The conclusion is that large wind farms (e.g., $n_{wt}$ = 500) can only be optimized with AD, as FD and CS would require CPU usage in the order of years.

### 4.1.2 Impact of Parallelization

Figure 6a shows the time per iteration for 1 CPU, 4 CPU, 16 CPU, and 32 CPU, whereas Figure 6b shows the speedup when comparing these different parallelization schemes. As Figures 5a and 5b demonstrated the superiority of AD, the simulations shown in Figures 6a and 6b used AD. The speedup computations consider 1 CPU as the baseline for comparisons. When several multi-starts are needed, our results show top-level parallelization of each optimization to be more efficient than parallelization of flow cases. According to Figures 6a and 6b, parallelization has a positive contribution to reducing $t_{iter}$ and increasing

the speedup. However, the speedup keeps constant with the $n_{wt}$. Moreover, the speedup when considering 4 CPUs against 1 CPU (Figure 6b) is four. The same comparison for 16 CPUs against 1 CPU results in a speedup of around 12, whereas 32 CPUs against 1 CPU have a speedup of around 16. These results indicate that the speedup achieved by parallelizing the flow

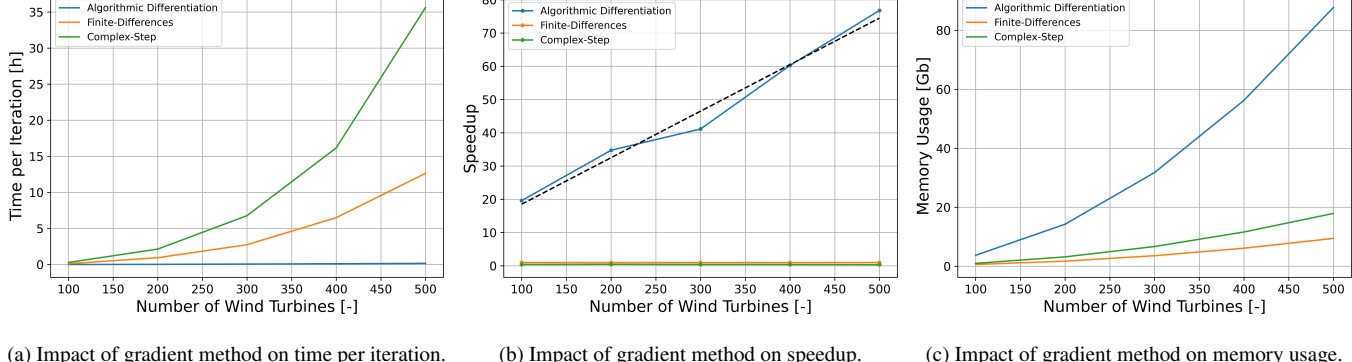

(a) Impact of gradient method on time per iteration.  (b) Impact of gradient method on speedup.  (c) Impact of gradient method on memory usage.

**Figure 5.** Impact of different gradient computation methods on time per iteration, speedup, and memory usage.

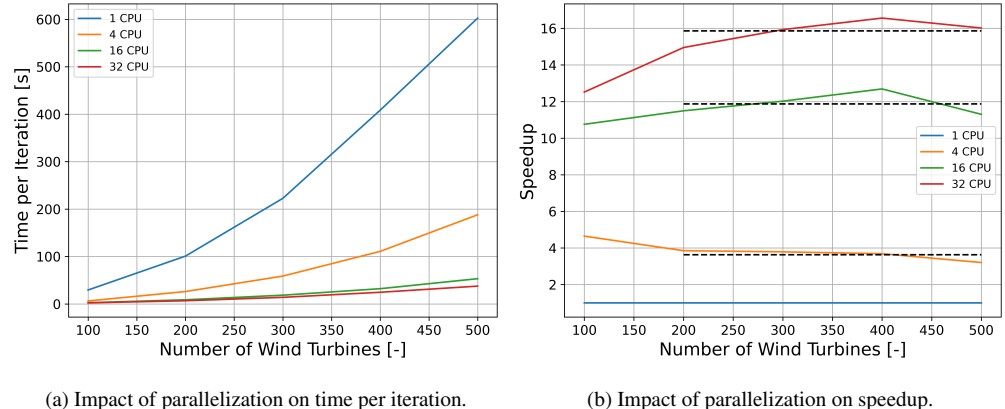

(a) Impact of parallelization on time per iteration.  (b) Impact of parallelization on speedup.

**Figure 6.** Impact of parallelization of the flow cases on time per iteration and speedup during GBWFLO.

cases does not linearly increase with the number of CPUs. These results show that parallelization of the multi-start process simulating one seed in one CPU seems more effective than parallelization of the flow cases. This means that if one needs to run hundreds of multi-starts, then better CPU utilization can be achieved by running each multi-start optimization in parallel, i.e. one optimization per CPU. This was confirmed by a small test where 100 multi-starts were optimized. In this test, the flow case parallelization approach was around two times slower than the multi-start parallelization approach.

### 4.1.3 Impact of Spacing Constraints

Figure 7 shows how the spacing constraint impacts the $t_{iter}$, indicating that handling spacing constraints does not scale well with $n_{wt}$. In the blue curve of Figure 7, each pair of turbines has an associated minimal spacing that must be satisfied, while the orange line has no pair-spacing constraints. Calculating the spacing between the wind turbines and the associated gradients is relatively fast. The bottleneck is the time spend on handling the constraints inside the optimizer, which is seen to be considerable



for large farms. In this example, handling the spacing constraint of each wind turbine pair in a setup with 500 wind turbines takes roughly two hours which slows down the iteration time by around 10 times. Obviously, wind turbines must be placed with

more than 1D spacing to avoid a collision, but this minimal distance is implicitly achieved even without spacing constraints in all optimizations performed in this study, see section 4.5 where also other issues related to too close spacing are discussed.

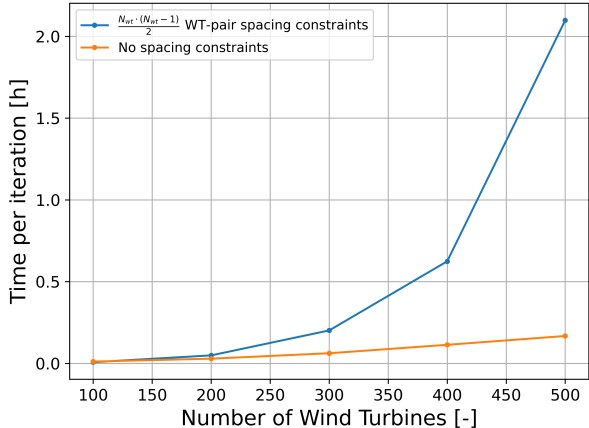

**Figure 7.** Influence of wind turbine pair-spacing constraints on time per iteration during GBWFLO.

## 4.2    Number of Iterations

The results in this section are based on optimizations of the faster scaled IEA 37 case study, described in section 3.2.

Figure 8 shows the $n_{iter}$ to achieve convergence as a function of $n_{wt}$ based on 5,000 optimizations of each farm size, $n_{wt} =$

(16, 36, 64, 130, 279). The mean $n_{iter}$ is seen to scale linearly with $n_{wt}$, $n_{iter} = 2.3n_{wt} + 16$. Additionally, two optimizations were performed with 566 wind turbines to verify the linear extrapolation to larger wind farm sizes.

In some cases $n_{iter}$ is considerably higher than the mean. We suspect that these outliers represent cases where the optimizer gets stuck in local minima.

In this case, $n_{iter}$ scales almost perfectly linearly with 2.3 times $n_{wt}$, but in general we expect $n_{iter}$ to be highly dependent

on the optimizer, its settings, e.g. tolerance, the nature of the objective function and the constraints, e.g. the shape of the boundary, as well as the scaling of the input, the objective function and the constraints. More investigation is needed to make a general conclusion.

The 5,000 optimizations of each farm size were performed with different levels of randomness, $random_{pct} = $ (0, 1, 10, 50, 100), but it was found that the amount of randomness only has a minor impact on the number of iterations, see Figure E2 in

Appendix E.

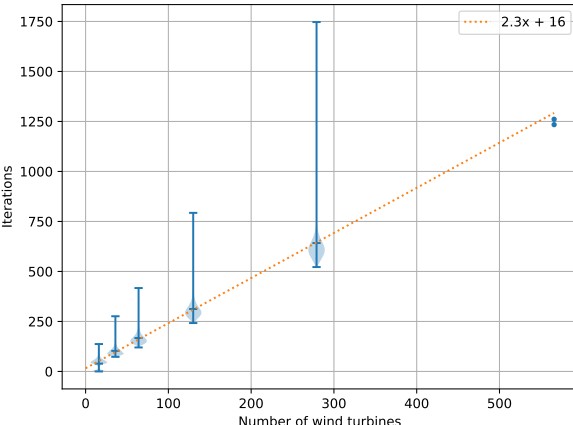

**Figure 8.** Number of iterations as a function of the number of wind turbines to achieve convergence. The site considered is the scaled IEA 37.

## 4.3 Number of Initial Starts

The results on reducing the number of initial starts presented in this section are based on simulations of the scaled IEA 37 study case described in section 3.2. Our approach used a heuristic algorithm to improve the multi-start GBWFLO optimization by providing a better guess for the initial layout. Section 3.4.1 described how the SMAST algorithm sequentially places turbines

in a gridded physical domain to obtain an initial layout. The SMAST algorithm evaluates the wind resource of each grid cell before placing each new turbine. The more refined the grid is, the more expensive it is to run SMAST. Before studying how to improve the multi-start, we ran a batch of simulations to check the sensitivity of the SMAST to several metrics (Appendix D and E, Figures D3, D4, E1, E2). The initial AEP provided by the SMAST increases as the grid resolution gets finer (as mentioned in section 3.4.1); however, there is a limit upon which the AEP no longer increases significantly (Figures D3 and D4). Based

on the results, the SMAST grid resolution adopted in this study is 3R (i.e., 3 times the wind turbine radius) for all the cases, except the 566 wind turbines case, where 5R was adopted to prevent memory problems. These resolution values provide, at the same time, suitable initial AEP at reasonable computational expenses. To get a sense of SMAST computational time for large wind farms, Table 2 shows a summary of the mean time of SMAST and the AEP gain, which is the percentage of improvement comparing the initial mean AEP provided by SMAST with $random_{pct} = 0$ and $random_{pct} = 100$. The computational time

for the smallest case $n_{wt} = 16$ is not high (only 3s) but it does not scale well and gets up to around 3 hours for the largest case in which $n_{wt} = 566$.

Figure 9 shows how the optimized AEP varies as a function of $n_{multistarts}$, considering different levels of randomness ($random_{pct}$) for the SMAST and different $n_{wt}$. The black dashed lines in each plot show the maximum AEP, and the gray bands a 99% confidence interval of the best result for the optimized AEP among two sets of 5,000 simulations with entirely

random initial layouts ($random_{pct} = 100$). In the 16 wind turbine case, Figure 9a, the random approach is equal or superior to





**Table 2.** SMAST Computational Time and Gain for the considered IEA37 cases

| $n_{wt}$ | 16 | 36 | 64 | 130 | 279 | 566 |
|---|---|---|---|---|---|---|
| $\overline{t_{init}}$ [s] | 3 | 11 | 53 | 413 | 3233 | 10742 |
| AEP Gain [%] | 9.07 | 11.23 | 12.14 | 11.52 | 11.05 | 10.54 |

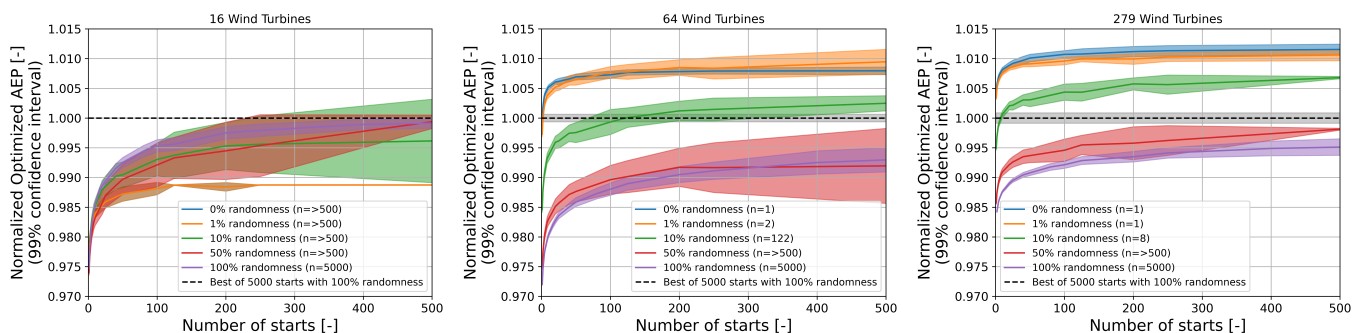

(a) Normalized Optimized AEP, 16 wind turbines.   (b) Normalized Optimized AEP, 64 wind turbines.   (c) Normalized Optimized AEP, 279 wind turbines.

**Figure 9.** Normalized Optimized AEP as a function of the number of initial starts, considering the IEA 37 study-case.

SMAST for all levels of randomness. In the 64 wind turbine case, Figure 9b, SMAST with 0% randomness needs only one start to obtain an AEP result that is as high as the best of 5,000 multi-start optimizations with 100% randomness. The SMAST with 1% and 10% require 2 and 122 initial starts (2-3 and 53-414, respectively, within a 99% confidence interval), respectively, to surpass the 100% random case with 5,000 starts. It is also seen that for more than 80 starts, 1% randomness improves the AEP

result. The 279 wind turbines case, Figure 9c, shows that the 0% and 1% SMAST cases need one initial start to surpass the 100% random case, whereas the SMAST 10% case requires approximately eight initial starts (4-16 within a 99% confidence interval). Note, however, that even though one start with 0% randomness is enough to surpass the random case, then it may still be beneficial to run with multiple starts. In this example, the maximum AEP can be increased by approximately 0.5% by running 30 starts instead of one.

These results indicate that more randomness gives higher AEP for a sufficiently high number of starts - the larger farms and the more randomness the more starts are needed. For the small $n_{wt} = 16$ case, "sufficiently high" is less than 20 starts. For the medium $n_{wt} = 64$ case, 80 starts are enough for 1% randomness while more than 500 starts are needed when introducing more randomness. For the large 279 wind turbine farm, "sufficiently high" is far beyond 500 even for the 1% randomness case. In summary, SMAST significantly reduces the number of starts required to get a high AEP result compared to the random

approach when optimizing large wind farm layouts.



## 4.4 Total Computational Time

This section summarizes the potential reduction of the total optimization time, $t_{total}$, for a wind farm with 500 wind turbines by using all the strategies approached in this work. The $t_{total}$ is estimated using Equation 1 and combines the iteration time found with the more realistic Hornsrev1 setup with the number of iterations and multi-starts found with the faster IEA37 setup.

Assuming that the result on number of multi-starts for 279 wind turbines is similar for a wind farm with 500 wind turbines.

First, the time per iteration for 500 wind turbines using FD and one CPU without spacing constraints is 12.6h (see Figure 5a). Additionally, the time for computing and handling spacing constraints for 500 wind turbines was found to be 1.9h (see Figures 7 and 6a). Therefore, the total iteration time without the methods proposed in this work is 14.5h. The number of iterations for a 500 wind turbine case is estimated to be 1,166 using the linear equation found in section 4.2. In this case, we set the number

of random multi-starts to 5,000, which is not even enough to get a result as good as one optimization with SMAST, if the trend from Figure 9 continues up to 500 wind turbines. Hence, the total optimization time on one node with 32 CPUs is estimated to be $14.5h/iter \cdot 1,166iter \cdot 5,000starts/32CPUs \approx 300years$.

Applying the methods presented in this work, i.e. AD, flow case parallelization on 32 CPUs, and no spacing constraints, the time per iteration is reduced to 38s (see Figure 6a). Using SMAST, one optimization is expected to give higher AEP than the

approach described above. The time to generate an initial layout by SMAST is $\approx 3.3h$. Therefore, the total optimization time is $38s/iter \cdot 1,166iter + 3.3h \approx 15.6h$.

A slightly higher AEP can be obtained by running, e.g. 32 optimizations with SMAST. In this case, it is more efficient to parallelize the starts rather than the flow cases. The iteration time is thereby increased to 603s (see Figure 6a), and the total optimization time becomes $(603s/iter \cdot 1,166iter + 3.3h) \cdot 32starts/32CPUs \approx 199h$.

## 4.5 Discussion

When running GBWFLO for large wind farms, one critical aspect is the spacing constraint. As previous works in WFLO considered small to medium wind farms, this problem was not explicitly apparent. That is why this work applied spacing constraints only to generate the initial layout (by SMAST) and disregarded them for the remaining optimization. That considerably reduced the computational expenses to achieve the objectives of this study. This was only possible because the wind turbines

did not end up too close. The strategy adopted in this work relied on the inherent behavior of wind turbine wake models, which places turbines apart from each other to produce more AEP, guiding the optimizer driver towards separating them. As the objective of the work is to show the impact of different gradient computation methods, parallelization, and the capabilities of SMAST, we do not expect disregarding the constraints to affect the results. For all the optimizations, the turbines were at least 1.4D separated apart in the final optimized designs, which is lower than typical spacing distances. It is crucial, though, to

reinforce that placing turbines too close to each other can cause problems related to mechanical loads on wind turbine components. Additionally, the applied engineering wake models do not include a dedicated near-wake model and their behavior close to the turbine is therefore missing some physical aspects. Furthermore, the constraint handling problem may be solved by constraint aggregation, but initial investigations showed a negative impact on the optimized AEP. Therefore, we decided





to run the optimizations in this study without spacing constraints and leave the in-depth investigation of spacing constraint
aggregation for future work.

In this work, the driver used for the optimization is the open-source SLSQP. The choice for this driver has to do with the possibility of a fully open-source simulation tool (even though other tools such as IPOPT could have been used). PyWake and TOPFARM, as mentioned in section 3, are open-source packages developed by DTU and well coupled with SLSQP. Additionally, SLSQP is currently commercially used by the wind energy industry to perform GBWFLO. Literature shows, though, some studies pointing towards the superiority of SNOPT for optimal AEP (Baker et al., 2019; Thomas et al., 2022b). Therefore, future work could use SNOPT or another driver to confirm the trends found in this work.

Another possibility to speed up wind farm optimization is considering a subset of the flow cases. For wind farm AEP computations, Thomas et al. (2022b) demonstrated that at least 40 or 50 wind sectors are necessary to run WFLO. However, wriggles can occur when simulating too few wind directions. In simple cases, the occurrence of such wriggles has been found to drastically increase the number of local minima. Taking this into consideration, in this work, we took the conservative approach of considering wind direction bins of 1°. However, we acknowledge other possibilities and intend to explore them in future work.

## 5    Conclusions

In this work, different strategies to accelerate WFLO of large wind farms with hundreds of turbines has been explored. We have focused on GB approaches as GF methods tend to scale poorly for problems with many design variables. We have separated the problem into reducing the iteration time, the number of iterations and the number of multi-starts (optimization with different initial layouts).

The time per iteration has been investigated using a realistic setup with scaled versions of the Horns Rev I wind farm (100 - 500 wind turbines). It was found that the iteration time can be decreased by computing gradients via AD compared to FD and CS. The speedup scales linearly with the number of wind turbines and was found to be around 75 times for a wind farm with 500 wind turbines. However, on personal computers or for even larger farms, AD may become unfeasible due to its extensive memory requirements.

Simulating the different flow cases in parallel is another approach to reduce the iteration time, but the speedup was found to be roughly constant with the number of wind turbines. Moreover, top-level parallelization was found to be more efficient. In general, we therefore recommend using the available CPUs to parallelize multi-starts with different initial layouts instead of flow cases.

Requiring all pairs of wind turbines to be separated by some minimum distance introduces a considerable number of optimization constraints. The time used to handle these constraints by the applied SLSQP optimizer scales very badly with the number of wind turbines, dominating the iteration time of wind farms with 300 wind turbines or more. The problem may be solved by using another optimizer or by constraint aggregation, but in this study, we ran the optimizations without spacing constraints and observed that all pairs of wind turbines were separated by at least 1.4D in all optimizations.



The number of iterations needed to achieve convergence was investigated using a faster setup (IEA37 with up to 566 wind turbines). The mean number of iterations was found to scale linearly with the number of wind turbines times 2.3. This result is assumed to be highly dependent on the optimizer, its settings, the boundary shape and the scaling of the inputs, outputs and constraints. The heuristic Smart-Start (SMAST) approach, which was used to obtain better initial layouts, did not manage to reduce the number of iterations significantly.

The number of multi-starts, i.e. number of optimizations performed with different initial layouts to obtain a result close to the global maximum, was also investigated using the faster setup (IEA 37 with up to 279 wind turbines). The number of multi-starts needed to reach 99.9% of the best AEP from 10,000 optimizations was found to depend on the wind farm size: 500 starts for 16 wind turbines, 2500 starts for 64 wind turbines, and 4000 starts for 279 wind turbines. For the biggest wind farm, however, it is suspected that 10,000 optimizations were not enough to find the global maximum.

Comparing SMAST with different levels of randomness ($random_{pct}$ = 0 - 100) revealed that more randomness in the initial layouts gives higher AEP when optimizing small wind farms (16 wind turbines), while less randomness, i.e. better initial layouts, is superior for large farms (64 and 279 wind turbines). It was found that the AEP obtained from one optimization initialized with SMAST ($random_{pct}$ = 0), was higher than the best AEP of 5,000 optimizations initialized with random wind turbine positions ($random_{pct}$ = 100). It is expected that the superiority of SMAST will increase even further for larger wind farms.

The reduction in total optimization time is estimated, assuming that the result on the number of multi-starts for 279 wind turbines is similar for larger wind farms and that the number of starts and iterations found using the faster IEA37 setup can be combined with the iteration time from the more realistic Horns Rev I setup. It was estimated that running one optimization with SMAST, AD, flow case parallelization and without spacing constraints instead of 5,000 optimizations with random initial layouts, FD, spacing constraints and top-level parallelization, reduces the total optimization time from around 300 years to 15.6 hours while increasing the AEP.

We suggest future works on large WFLO to explore the effect of constraint aggregation methods on iteration time and optimized AEP, and to test the proposed approaches with other optimizers and wind farm setups to generalize the results of this present work.

*Code availability.* The code for the Smart-Start (SMAST) algorithm is available in PyWake, an open-source tool developed by the Technical University of Denmark.

*Author contributions.* RVR, MMP, JPSC, JQ, and PR developed the problem formulation and designed the experiments. RVR, MMP, and JPSC contributed to numerical simulations. RVR wrote the first draft. RVR, MMP, JPSC, JQ, and PR reviewed and edited the manuscript.



*Competing interests.* The authors declare no competing interests in this article.

*Acknowledgements.* The authors would like to thank the AIT department for providing access to the Sophia HPC Cluster at the Technical University of Denmark, DOI: 10.57940/FAFC-6M81.

## Appendix A: Acronyms

*AEP*: Annual Energy Production

*AD*: Algorithmic Differentiation

*SMAST*: Smart-Start

*GWEC*: Global Wind Energy Council

*WFLO*: Wind Farm Layout Optimization

*GF*: Gradient-Free

*GA*: Genetic Algorithm

*PSO*: Particle Swarm Optimization

*RS*: Random-Search

*GBWFLO*: Gradient-Based Wind Farm Layout Optimization

*L-BFGS*: Limited-memory Broyden-Fletcher-Goldfarb-Shanno

*SNOPT*: Sparse Nonlinear Optimizer

*FD*: Finite-Differences

*CS*: Complex-Step

*GB*: Gradient-Based

*SA*: Simulated Annealing

*IPM*: Interior Point Method

*SQP*: Sequential Quadratic Optimization

*BG*: Bastankhah Gaussian

*SBG*: Simple Bastankhah Gaussian

*SLSQP*: Sequential Least Squares Programming

*OpenMDAO*: Open Multidisciplinary Design, Analysis, and Optimization

*OOB*: Out of boundaries



**Appendix B: Nomenclature**

$t_{iter}$: Time per iteration

$n_{multistarts}$: Number of multi-starts

$n_{iter}$: The number of iterations until convergence

$t_{total}$: Total computational time for GB

$n_{wt}$: Total computational time for GB

$x$: Layout coordinate of the wind turbine in the x-direction

$y$: Layout coordinate of the wind turbine in the y-direction

$C_k$: Wind farm boundary constraint

$k$: Integer number to represent the index of $C$ and the turbine number

$d$: Wind direction

$u$: Wind speeds

$N_\theta$: number of wind direction

$N_u$: Number of wind speeds

$P_{d,u}$: Represents the power output of the wind farm given by the wind turbine coordinate vectors $x$ and $y$

$\rho_{d,u}$: Number of wind speeds

$R_{wf}$: Wind farm radius

$x^{UL}$: upper left coordinate that defines the parallelogram boundaries

$x^{UR}$: upper right coordinate that defines the parallelogram boundaries

$x^{LL}$: lower left coordinate that defines the parallelogram boundaries

$x^{LR}$: lower right coordinate that defines the parallelogram boundaries

$C_T$: Thrust coefficient

$WSBins$: Bins for the wind speeds

$WDBins$: Bins for the wind directions

$\mathcal{L}$: Vector of potential positions for wind turbines

$\alpha_{best}$: points in $\mathcal{L}$ with the highest AEP

$p$: point among $\alpha_{best}$

$OOB$: Points out of the boundaries of the wind farm

$P$: Wind turbine positions vector

$random_{pct}$: level of randomness for layouts generated by SMAST

$D$: Wind turbine diameter

$AEP_{ub}$: the upper bound of the AEP within a 99% confidence interval

$AEP_{lb}$: the lower bound of the AEP within a 99% confidence interval

$m$: m is the number of chunks of the 10,000 simulations with initial layouts randomly generated





$AEP_{mean}$: is the mean of all the maximum AEP values of each chunk m

$\sigma$: standard deviation of the maximum AEP values of all chunks of size m

## Appendix C:  Equations for Gradients Computations

Equation (C2) shows the FD generalized formula containing all the high-order truncation terms for the forward difference. Equation (C1) reduces to Equation (C2) when considering only the first-order truncation terms, where O(h) refers to the

truncation error.

$$f(x + h\widehat{e}_j) = f(x) + h\frac{\partial f}{\partial x_j} + \frac{h^2}{2!}\frac{\partial^2 f}{\partial x_j^2} + \frac{h^3}{3!}\frac{\partial^3 f}{\partial x_j^3}..., \tag{C1}$$

where h represents FD step size, $\widehat{e}_j$ is the unit vector at the $j^{th}$ direction, as shown by Martins and Ning (2021).

$$\frac{\partial f}{\partial x_j} = \frac{f(x + h\widehat{e}_j) - f(x)}{h} + O(h) \tag{C2}$$

where O(h) represents the truncation error.

Likewise to FD, CS Equation (C3) reduces to (C4) when disregarding higher order truncation terms.

$$f(x + ih\widehat{e}_j) = f(x) + ih\frac{\partial f}{\partial x_j} + \frac{h^2}{2!}\frac{\partial^2 f}{\partial x_j^2} + i\frac{h^3}{3!}\frac{\partial^3 f}{\partial x_j^3}..., \tag{C3}$$

$$\frac{\partial f}{\partial x_j} = \frac{Im(f(x + ih\widehat{e}_j))}{h} + O(h^2) \tag{C4}$$



## Appendix D: Grid Resolution


For the charts on the left on Figures D1 and D2, the AEP is calculated for a wind turbine with all potential positions taking into account wakes from the already added wind turbine(s) (in black color). A new wind turbine (red) is added at the best position. The chart on the right shows the final layout provided by SMAST, where all 64 wind turbines have been placed.

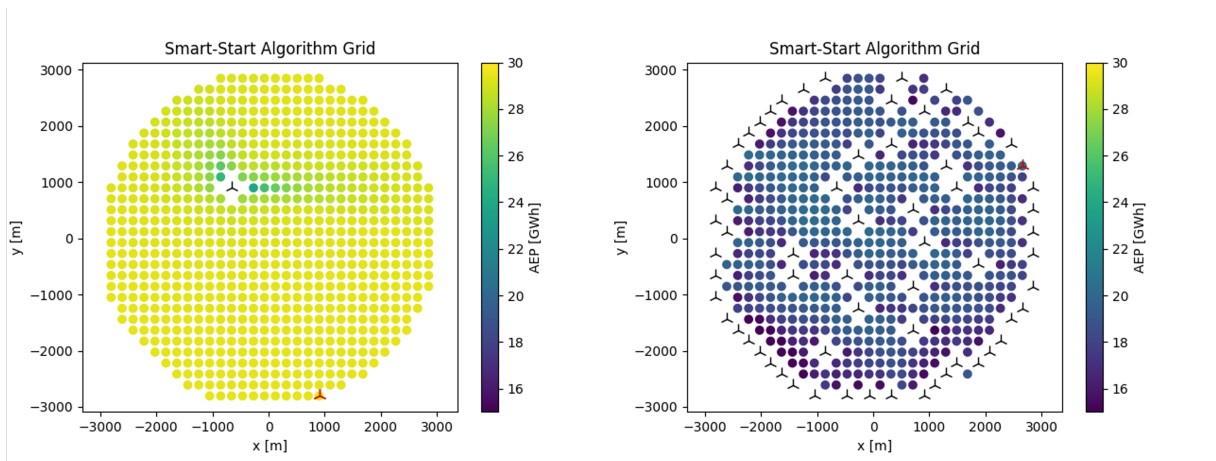

**Figure D1.** Example of a SMAST grid with 3R resolution for $n_{wt} = 64$.

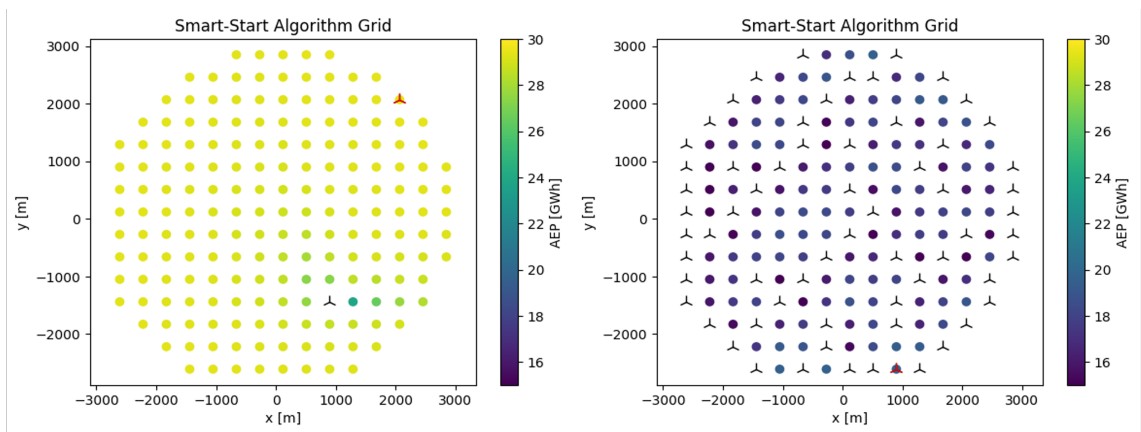

**Figure D2.** Example of a SMAST grid with 6R resolution for $n_{wt} = 64$.

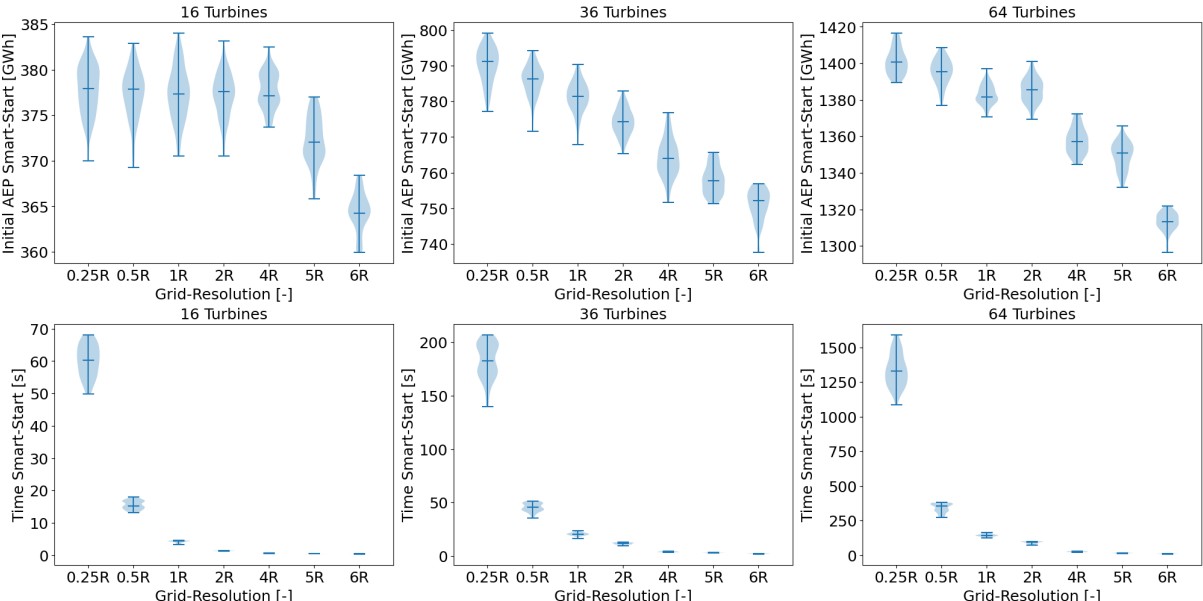

**Figure D3.** SMAST Grid resolution for the small/medium wind farms.

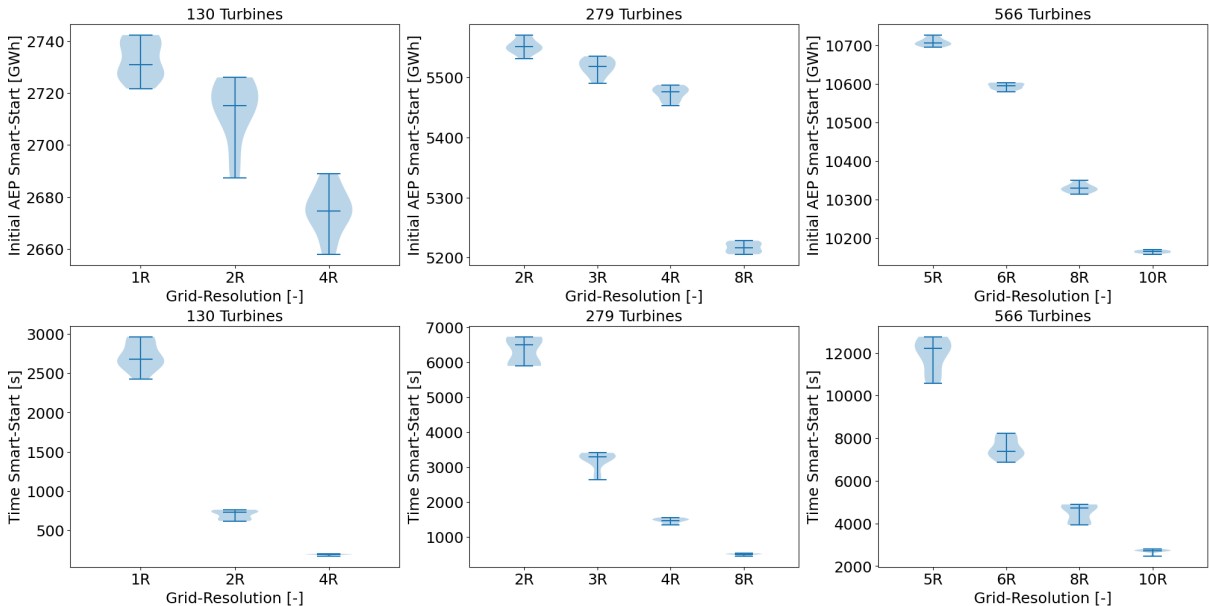

**Figure D4.** SMAST Grid resolution for the large wind farms.






## Appendix E: Random Parameter Impact

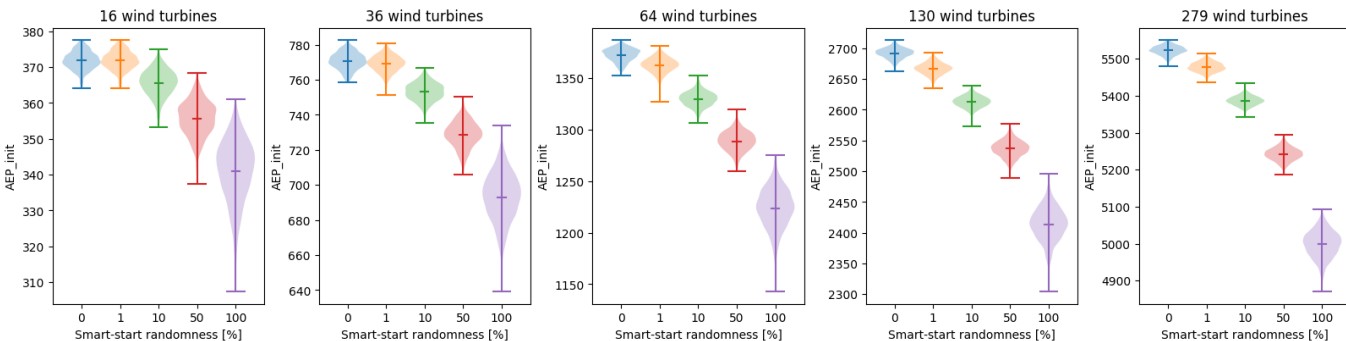

**Figure E1.** Impact of $random_{pct}$ on the initial AEP.

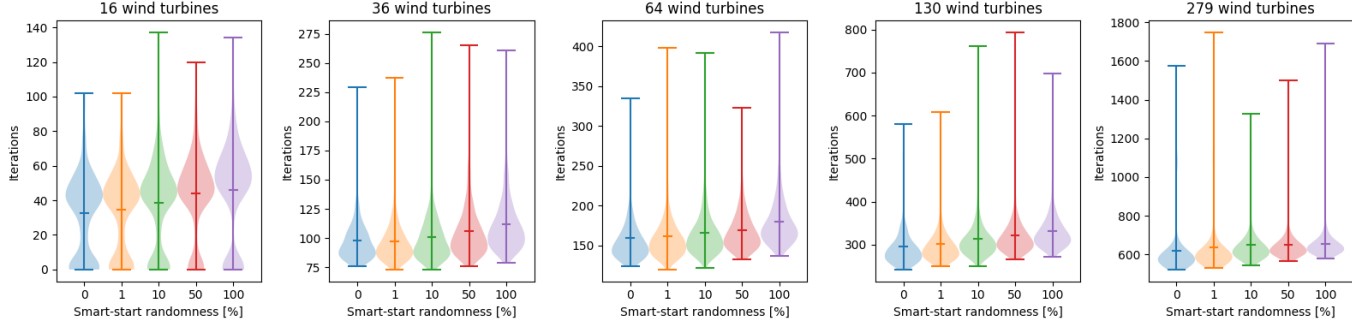

**Figure E2.** Impact of $random_{pct}$ on the number of iterations.





## Appendix F:  Final AEP and Optimization Time

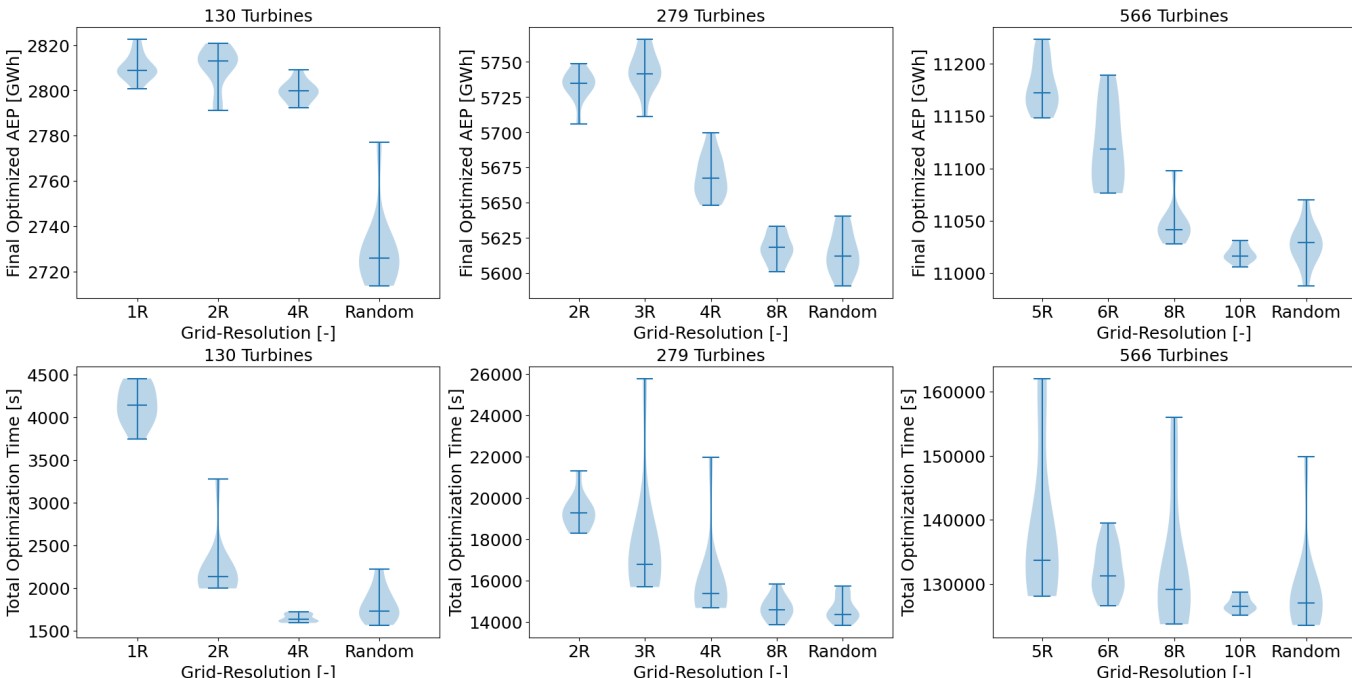

**Figure F1.** Final AEP and optimization time of SMAST as compared to randomly generated layouts.



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
