# Peer review of "Speeding up large wind farms layout optimization using gradients, parallelization, and a heuristic algorithm for the initial layout"

_Wind Energy Science, 2023_

## Author Comment (AC1)

**Review WES**

October 5, 2023

On behalf of all the authors, we would like to thank Reviewer 1 (Pietro Bortolloti) and the anonymous Reviewer 2 for reviewing our manuscript. Your suggestions significantly improved our manuscript, and we have attempted, to the best of our knowledge, to address all of them.

**We have used different colors throughout the review process:**

Green: The green color indicates one comment/question from a Reviewer

Black: The black color generally refers to content from the original submission of the manuscript (from the published preprint)

Blue: the blue color indicates either our answer or content added to the original text in the published preprint in WES

**Reviewer 1**

- Page 4 line 113: typo, reference is duplicated

Answer: Thank you for identifying the referred typo, we have fixed by excluding one of the occurrences of "Perez et al.".

- Page 7 line 191: how off are the AEP values?

Answer: The plots from Figure 1 (not included in the manuscript) show one of our earliest simulations. At that point, we hadn't done a fine-tuning to get a faster optimization workflow/setup. We considered 180 wind direction bins for comparisons at that time. The initial values of the AEP were around 1.5 to 3.5% different compared to the benchmark by Baker (2019). As a reference, the benchmark performed by Baker (2019) considered 16 wind directions. The final optimized AEP, however, was within around 7.5% and 8.5% different when comparing the 16 wind direction bins and the 180 wind direction bins.

In the end, we decided to be conservative and use 360 wind direction bins due to the occurrence of wriggles, as described in lines 406 to 412:

"Another possibility to speed up wind farm optimization is considering a subset of the flow cases. For wind farm AEP computations, Thomas et al. (2022b) demonstrated that at least 40 or 50 wind sectors are necessary to run WFLO. However, wriggles can occur when simulating too few wind directions. We define wriggles as direction-dependent variations in wind turbine wakes when averaging all the wind directions with their sector frequency weight. In simple cases, the occurrence of such wriggles has been found to drastically increase the number of local minima. Taking this into consideration, in this work, we took the conservative approach of considering wind direction bins of 1°. However, we acknowledge other possibilities and intend to explore them in future work."

[Figure]

Figure 1: Comparisons showing final optimized AEP according to different numbers of wind directions.

Aiming to improve the manuscript following the Reviewer 1 question, we have included the estimates mentioned here in line 191:

"Note that, in the current study, we simulate 360 wind directions (as shown in Table 1) while only 16 wind directions were considered in the original IEA Wind Task 37 case study. Our results are, therefore, not directly comparable to the AEP results of the benchmark in Baker's work (Baker et al., 2019). The final optimized AEP values from Baker's benchmark were around 7.5% to 8.5% higher than our preliminary results."

- Page 7 line 180: please add "wind farm" to the Horns Rev I
Answer: Thank you for your comment. We have added "wind farm" to line 180. Now, it reads:

"The simulations to investigate the time per iteration were performed with a realistic setup, the Horns Rev I wind farm. "

- Figure 1b reports a Ct for the IEA 3.4 that is clearly wrong. Not sure where the value comes from, but this file might be a better source for Ct

Answer: The IEA 37 wind turbine we used in this work is slightly different from the IEA 3.4MW reported in:
**Bortolotti, P., Tarres, H. C., Dykes, K. L., Merz, K., Sethuraman, L., Verelst, D., and Zahle, F. (2019). IEA Wind TCP Task 37: Systems engineering in wind energy-WP2. 1 Reference wind turbines (No. NREL/TP-5000-73492). National Renewable Energy Lab.(NREL), Golden, CO (United States).**

The wind turbine we used in this study is reported in:

**International Energy Agency. 2019. IEA WIND TASK 37 on Systems Engineering in Wind Energy (2019): Wake Model Description for Optimization Only Case Study. Tech. rep., International Energy Agency, Available in: https://github. com/byuflowlab/iea37-wflo-casestudies/blob/master/cs1-2/iea37-wakemodel. pdf (last access: 03 October 2023).**

The same turbine was later used in the benchmark study below:

**Baker, N. F., Stanley, A. P., Thomas, J. J., Ning, A., and Dykes, K. (2019). Best practices for wake model and optimization algorithm selection in wind farm layout optimization. In AIAA Scitech 2019 forum (p. 0540).**

This is a 3.35MW idealized wind turbine, and as reported in the references above, a constant $C_T$=8/9. Initially, the reasoning behind this choice was the possibility of verification of our model in PyWake with the benchmark provided in the study. Given the same conditions, e.g., the number of wind speeds (only 9.8m/s) and wind directions (16 sectors), the PyWake results for the AEP match exactly with the results from the benchmark study. We have named this simplified and, therefore, faster wake model as "Simple Bastankhah Gaussian" as it is based on the idealization described here.

In order to clarify readers and Reviewer 1, we have changed lines 178-185 as below:

"The results for the number of iterations and number of multi-starts are based on more than 55,000 GBWFLOs, and these were performed with a faster setup, which uses  idealized 3.35MW wind turbines with constant $C_T$, site, and wake model definitions from the IEA Wind Task 37 case study 1 (IEA Wind Task 37, 2018; Baker et al., 2019). These wind turbines are slightly different than the reference IEA 3.4MW wind turbines defined by Bortolotti et al., 2019 . Extra cases were designed to scale the analysis (Figure 3). In this simplified setup, only the rated wind speed (9.8m/s) is simulated, and all wind turbines operate with constant  $C_T \approx 8/9$ ."

- Figure 2: I don't understand why 80 turbines are marked in black and others in white

Answer: The 80 turbines marked in black are the original turbines from the Horns Rev I layout, whereas the turbines marked in white represent the rows and columns added to the original layout.

In order to clarify the reviewer and readers, we have changed the caption of Figure 2 including:

"**Figure 2**. Considered variations of the scaled Horns Rev I site. The 80 turbines marked in black are the original turbines from the Horns Rev I layout, whereas the turbines marked in white represent the rows and columns added to the original layout."

- Page 8 line 203: the sentence seems a little clunky

Answer: We have updated the sentence in line 203 to:

"This work implemented parallelization in a computational cluster. Single node operation was utilized with each node being composed of 2x AMD EPYC 7351 16 core CPUs, @2.9GHz, with 128GB of RAM."

- Page 10 line 218: what does AEP best mean?
Answer: As also pointed out by Reviewer 2, there was a typo in that symbol. The typo happened because of a mistake in the LATEX file. The arrow symbol was supposed to be the symbol $\mathcal{L}$ and should read as below:
"Next, SMAST randomly selects a point p among the points associated with the highest AEP, $\mathcal{L}_{best}$, and places the next turbine at p."

$\mathcal{L}_{best}$ is the an array with potential positions with turbines, $\mathcal{L}_{best}$ is the same type of array but with the positions with the highest AEP.

---

## Author Comment (AC2)

**Review WES**

October 5, 2023

On behalf of all the authors, we would like to thank Reviewer 1 (Pietro Bortolloti) and the anonymous Reviewer 2 for reviewing our manuscript. Your suggestions significantly improved our manuscript, and we have attempted, to the best of our knowledge, to address all of them.

**We have used different colors throughout the review process:**

Green: The green color indicates one comment/question from a Reviewer

Black: The black color generally refers to content from the original submission of the manuscript (from the published preprint)

Blue: the blue color indicates either our answer or content added to the original text in the published preprint in WES

**Reviewer 2**
1) Technical Comments

- Line 67: Can the need for multiple initial starts be discussed in the introduction? It is a specific issue for gradient-based optimization and is an important background for one of the paper's objectives.

Answer: We have added content to the introduction to introduce how GBWFLO is susceptible to getting stuck in local minima. The paragraph in lines 30-44 has been changed as below. In the original paragraph, Wind Farm Gradient-Free Optimization is introduced, and some problems are described. Then, Gradient-Based Wind Farm Layout Optimization (GBWFLO) is introduced. Now, with the corrections suggested by Reviewer 2, we introduced the problem of Gradient-Based Optimization getting stuck in local minima, and briefly mention that efficient multi-start can help overcome the problem:

"Since the first work on WFLO by Mosetti et al. (1994) using a Gradient-Free (GF) approach, the literature on the topic massively evolved around GF. GF-based approaches on the topic include metaheuristic methods such as Genetic Algorithm (GA) (González et al., 2018; Wang et al., 2015; Parada et al., 2017), Particle Swarm Optimization (PSO) (Hou et al., 2016; Pillai et al., 2017; Veeramachaneni et al., 2012; Wan et al., 2012; Pookpunt and Ongsakul, 2016), Random-Search (RS) (Feng and Shen, 2017b, a), and many others. GF methods explore the entire design space and may find the global optimum at some point, but processing time explodes with the number of design variables. Therefore, GF methods tend not to scale well for problems with many design variables (Martins and Ning, 2021; Ning et al., 2019) and are more suitable for smaller problems (Wright et al., 1999). Gradient-Based Wind Farm Layout Optimization (GBWFLO) has been explored more since a few years ago. Research in the field has been evolving, including analytical computation of the gradients (Guirguis et al., 2016, 2017; Stanley et al., 2019), a quasi-Newton limited-memory optimizer called Limited-memory Broyden-Fletcher-Goldfarb-Shanno (L-BFGS) that estimates the inverse of the Hessian Matrix using a generalized secant method (van Dijk et al., 2017; Croonen- broeck and Hennecke, 2021), another limited-memory optimizer called SNOPT (Sparse Nonlinear Optimizer) that explores the sparsity of the Jacobian matrix (Tingey and Ning, 2017), SNOPT with Finite Differences (FD)

(Fleming et al., 2016), SNOPT with analytical gradients (Gebraad et al., 2017), and Adjoints (King et al., 2017; Allen et al., 2020). Mittal et al. (2016) developed a hybrid GF (GA) and GB (fmincon) algorithm. A concern in GBWFLO is getting stuck in local minima due to the multi-modality of the problem, as visually demonstrated in the literature (Thomas et al., 2022b). One possible strategy to overcome local minima is to perform multi-starts by running multiple optimizations with different initial solutions. Multi-start GBWFLO can explore the design space more broadly, avoiding potential non-optimal final solutions. However, the extra costs to run multi-starts can become another concern for larger problems. In this context, efficient multi-start is potentially a way of speeding up GBWFLO."

- Table 1: What is the reasoning for choosing 500 turbines as the "large wind farm" size? Is this the trend in expected wind plant sizes in the coming decades? It is obviously a much larger number of turbines than modern large wind farms of 100 turbines. I highlight this point because the main conclusions of the article are based on comparing optimization performance on this large 500 turbine farm, and the results are more modest for the 100-300 turbine range.

Answer:
The motivation for writing the article came from our interaction with the wind energy industry. The reason for choosing 500 turbines was driven by an industry request in a research project with a major leader Danish company we worked on. They brought us the challenge of optimizing the layout of 500 turbines in 6 hours, and we attempted to solve the problem using the various strategies described in this article. Our industry partner knew that optimization techniques for more modest wind farms (e.g., 100 wind turbines) might fail when we scale the problem to larger clusters. Another example of a larger wind farm cluster is the energy islands concept. Even though composed of several normal-sized farms, they could be seen as one bigger cluster of turbines. We have slightly altered the first paragraph in line 26, adding:

"The use of wind energy worldwide increases year by year. The global cumulative wind power capacity reached 837GW by the end of 2021, with a prediction of around 3,200GW by 2030 (GWEC, 2022). This growth opens the path for the wind energy industry to consider building larger wind farms."

All the results in this article were based on estimates for each of the terms in Equation (1), where:

- $t_{iter}$: based on the Horns Rev I setup with wind speeds from 3 to 25m/s (bins of 1m/s) and wind direction from 0 to 360° (bins of 1°)

- $n_{iter}$: based on the statistical fit from Figure 8, where we performed thousands of simulations for the IEA 37 setup and two simulations to confirm the trend for 566 wind turbines.

- $t_{init}$: estimated using Smart-Start (SMAST). Figures D3 and D4 show estimates for the time to get initial layouts (e.g.,$t_{init}$) using SMAST for various grid sizes.

- $n_{multistarts}$: based on SMAST algorithm, simulations presented in Figure 9

- $n_{cpu}$: estimates done for each of the layouts from Figure 2 (realistic Horns Rev I setup), and extrapolation within each interval (e.g., from $n_{wt}$=100 to $n_{wt}$=200, and so on). Results for each layout are shown in Figure 6.

The estimates for the total computational time ($t_{total}$) made in section 4.4, along with the conclusions drawn in the manuscript, took into consideration several wind farm sizes up to $n_{w}t$=500 or $n_{w}t$=566. The procedures that extrapolated results for a $n_{wt}$=300 to a $n_{wt}$=500 were: 1) Estimating $n_{iter}$: we confirmed the trend observed in Figure 8 with two extra simulations for the case with $n_{wt}$=500; 2) Estimating $n_{multistarts}$: In section 4.4, we did reinforce that our results are valid assuming the trends for $n_{wt} = 279$ hold for $n_{wt} = 566$.

Line 350: "Assuming that the result on a number of multi-starts for 279 wind turbines is similar for a wind farm with 500 wind turbines."

This assumption was made by analyzing the trends observed in Figure 9: only two and one simulations were required for $n_{wt}$=64 and $n_{wt}$=279, respectively, to surpass optimized AEPs with initial layouts randomly generated. Based on these results, we expect only one SMAST-generated layout for $n_{wt}$=500 to surpass a simulation with a randomly initialized layout of the same size.

- Line 185: Is this the correct Ct value? It seems extremely high for the rated wind speed. I'm not sure if the results would change much, but I expect that the wake interactions are much larger than they would otherwise be, which may affect convergence. Also, why was a constant Ct value even necessary? Why does the Cp value not need to be constant?

Answer: There is a typo in the $C_T$ value reported in line 185. The correct is $C_T \approx 8/9$. The IEA37 wake model (here called Simple Bastankhah Gaussian), however, is formulated such that the wake width at the rotor ($\epsilon$) equals the Bastankhah Gaussian when using $C_T \approx 0.964$ to calculate $\beta$. This formulation is likely in that way to avoid the constant wake depth in the near wake. //
All the references for the derivations of the IEA37 wake model have been included in the article, including:
1) The benchmark reference already included in the original submission:
Baker, N. F., Stanley, A. P., Thomas, J. J., Ning, A., and Dykes, K. (2019). Best practices for wake model and optimization algorithm selection in wind farm layout optimization. In AIAA Scitech 2019 forum (p. 0540).
2) One more reference included in the revision to show the derivations of the IEA 37 wake model:
International Energy Agency. 2019. IEA WIND TASK 37 on Systems Engineering in Wind Energy (2019): Wake Model Description for Optimization Only Case Study. Tech. rep., International Energy Agency, Available in: `https://github.com/byuflowlab/iea37-wflo-casestudies/blob/master/cs1-2/iea37-wakemodel.pdf(lastaccess:03October2023).`

In regards to the $C_P$, we do not make any mention of it in the text. In the case of the IEA 37 wind turbine, the $C_P$ would be constant due to the setup using only one wind speed. The V80 would have a variable $C_P$.

- Line 231: Is the minimum spacing between turbines in the optimized layout sensitive to the resolution of the grid spacing in the SMAST initial layout? Is there a resolution of the initial layout that would lead to issues when the spacing constraint is omitted from the problem?

Answer: We do not expect the grid resolution of the initial layout would lead to the issues referred by the reviewer, as the optimizer moves the turbines apart to avoid the highest wake regions. We have added a sentence in line 235:

"Another parameter that influences the AEP provided by the SMAST is the 230 resolution of the grid defined in Algorithm 1. Figures D1 and D2 show examples of SMAST AEP flow maps of the potential positions L with grids with resolutions of 3R and 6R (i.e. the distance between points in L), respectively. Figures D3 and D4 show how the AEP and the computational time vary according to the grid size. The finer the SMAST grid, the higher the AEP (except for nwt = 16 where it stabilizes after 4R) but also the higher the computational time of SMAST. Memory can also be a problem in running SMAST if the grid is too refined. We do not expect the minimum spacing between turbines in the optimized layout to be sensitive to the grid resolution provided by SMAST, as the optimizer moves the turbines apart to avoid the highest wake regions.

- Line 292: Why is "handling" the constraint so expensive compared to the calculation of the relative spacing?

Answer: We used SLSQP (Sequential Least Squares Quadratic Programming) for optimizations in this work. Calculating the distances between 500 points is very fast – it takes around 0.01s, and calculating the gradients is similarly fast. SLSQP, however, needs to compute the Lagrangian multiplier for all constraints, and we assume that is what takes most of the almost two hours that the iteration time increases when

introducing WT-pair spacing constraints.

"Figure 7 shows how the spacing constraint impacts the $t_{iter}$ , indicating that handling spacing constraints does not scale well with $n_{wt}$. In the blue curve of Figure 7, each pair of turbines has an associated minimal spacing that must be satisfied, while the orange line has no pair-spacing constraints. Calculating the spacing between the wind turbines and the associated gradients is relatively fast. The bottleneck is the time spent on handling the constraints inside the optimizer, which is seen to be considerable for large farms. The optimizer used in this work is the SLSQP. Calculating the distances between 500 points takes around 0.01s, and calculating the gradients is similarly fast. SLSQP, however, needs to compute the Lagrangian multiplier for all the constraints, and it is assumed is what takes most of the almost two hours increase in the iteration time (Figure 7) when introducing WT-pair spacing constraints. In this example, handling the spacing constraint of each wind turbine pair in a setup with 500 wind turbines takes roughly two hours which slows down the iteration time by around 10 times. Obviously, wind turbines must be placed with more than 1D spacing to avoid a collision, but this minimal distance is implicitly achieved even without spacing constraints in all optimizations performed in this study, see section 4.5 where also other issues related to too close spacing are discussed.

- Figure 7: Is the spacing constraint scaling with $n_{turb}^2$? That is the rough scaling you would expect from the constraint definition.
Answer: Yes, the calculation of the distances and their gradients scales with $n_{wt}^2$, but in this case, the calculation only comprises a small fraction of the total time. At the same time, the major part is spent in SLSQP, where it is less obvious how the running time scales with the number of constraints. Please, check:
    https://github.com/scipy/scipy/blob/main/scipy/optimize/slsqp/slsqp_optmz.f

- Line 300: Does this linear relation hold for the median number of iterations? The median is less susceptible to outliers and is arguably a more useful result for gauging an expected number of iterations.

Answer: Yes, the results are very similar. The linear fit of the median has the same slope as we use in Figure 8, with just one decimal difference. We have added that observation in line 300:

"Figure 8 shows the $n_{iter}$ to achieve convergence as a function of $n_{wt}$ based on 5,000 optimizations of each farm size, $n_{wt} = (16, 36, 64, 130, 279)$. The mean $n_{iter}$ is seen to scale linearly with $n_{wt}$, $n_{iter} = 2.3 n_w t + 16$. The linear fit of the median has the same slope as Figure 8 with just one decimal difference. Additionally, two optimizations were performed with 566 wind turbines to verify the linear extrapolation to larger wind farm sizes. "

- Table 2: What is the initialization time for the random layouts? For 566 turbines, I expect the time would be non-negligible to simply satisfy spacing and boundary constraints. Perhaps the factor of time between the two degrees of randomness could be given if the authors wish to put the SMAST initialization in context?

Answer:
The initialization time to generate random layouts is nearly negligible compared to, for instance, SMAST with $random_{pct} = 0$, as the expensive loop to compute AEP and store information about remaining cells in each iteration is bypassed. Satisfying spacing constraints, as well as physical boundary constraints, is done after placing each new turbine by excluding neighboring cells not fulfilling proximity and out of the domain. This operation is not expensive. We have included Figure 1 in the appendix of the article as a reference.

Line 316: Is this tuning always required when using the SMAST algorithm? It seems like there is a non-trivial amount of tuning that would increase development/design time compared with the simple random layout generation.

Answer: We have attempted to show in Figures D3 and D4 the grid resolution relationship with the initial AEP provided by SMAST. Those results could be a reference for readers to offer insights on choosing

[Figure]

Figure 1: Smart-Start initialization time.

a suitable grid with good AEP results rather than tuning every time. The essential idea to bear in mind is what we reinforced in lines 318 and 319:

"The initial AEP provided by the SMAST increases as the grid resolution gets finer (as mentioned in section 3.4.1); however, there is a limit upon which the AEP no longer increases significantly (Figures D3 and D4)."

- Figure 9: Why would different levels of randomness in the initial layout result in higher optimal AEP for different sized wind farms? For example, more randomness is resulting in higher optimal AEP for 16 turbines, while practically no randomness is best for the 64 and 279 turbine farms. Is there some numerical issue in the modeling of the small wind farms or in the optimizer that is made up for with more randomization of the initial layout? Are there more local optima for the small farm layout?

Answer: The authors do not expect numerical issues in modeling small wind farms. When the number of wind turbines increases, a larger portion of the area is affected by wake effects. Consequently, using an algorithm that evaluates each cell before placing a new turbine makes more sense. In the case of smaller wind farms, there are fewer cells in the physical domain. Therefore, choosing where to place a new turbine is less critical compared to the case with more turbines.

Moreover, a randomness value ($random_{pct}$) lower than 100% means that the initial layout is limited to a subset of the design space. If the optimizer is not able to escape the local minima, this may also limit the solution space. For small wind farms, a higher AEP may be found by a random guess in the solution space that is not accessible when using less randomness in the initial start. For larger wind farms, finding a better solution by random is not realistic.

- Figure 9: Between 200-5000 multistarts, there is only an increase in AEP of about 1%. With this low-fidelity wake model for AEP, can you even be confident in this 1% change? The cost of the optimization study increases by a factor of 25 if you insist on getting an extra 1% AEP out of the study, but that may not even be a real improvement if you validated with high-fidelity simulations.

Answer: Indeed, we cannot say we are confident that 1% AEP improvement given by the low-fidelity wake models used results in 1% higher AEP in high-fidelity wake models. We are also not confident that a 1% improvement in high-fidelity models would result in 1% in real life.

Moreover, it is likely not worth using 25 times more to get the 1% higher AEP given by this low-fidelity wake model. Our intention with Figure 9 is to:

- show how convergence behaves

- show that it may be worth running extra 10 or 100 multi-starts

• show that less randomness is superior for large wind farms

- Line 337: Why would 0% randomness require any multistarts? If there is no randomness, would you not arrive at the same optimal AEP each time?

Answer: Section 3.4.1 provides a detailed explanation of the Smart-Start (SMAST) algorithm. If SMAST is run without randomness ($random_{pct} = 0$), the algorithm selects the point $p$ with the highest AEP among the $\alpha_{best}$ points. If multiple points provide the highest AEP, e.g., in the first iteration assuming a uniform site, SMAST selects one of these points randomly. That random choice causes the cases with $random_{pct}$=0 (i.e., 0% randomness) not to arrive at the same optimal AEP each time, consequently justifying the need for multi-starts.

-Line 358: Why was flow case parallelization done if you argued against it in Section 4.1.2? Is it because you are only doing one SMAST start, and so you can utilize the 32 CPUs on the node without penalty?

Answer: We studied parallelization of the flow cases up to section 4.1.2, and then again we described this option in section 4.4 during the total time discussion. Even though it is not an optimal utilization of your computational resources, you will get results faster by parallelizing the flow cases in the following cases:

• If you have enough computational resources

• If you only want one initial start, for instance, if you do not believe in the percentage of improvement due to multi-starts

**Typographical Comments**

Line 82: "The CS method typically doubles computation time, as there are two times more bits in each value." What is meant by the last part of this sentence? Is it that the complex step requires operations with real and imaginary values (as opposed to just real), so the number of floating point operations in the Taylor expansion are roughly doubled?

Answer: Yes. As shown in Appendix C4, we need to add a complex step to an input and run $f(x + ih\widehat{e}_j)$ to find the gradient with respect to that input. This means that all operations in f, which depends on x, need to be executed on both the real and the imaginary part. We have changed and added explanations to line 83:

"Computing gradients can be done with different methods. In Algorithmic Differentiation (AD), all the lines of code are differentiated. These lines are usually composed of simple mathematical operations. AD performs differentiation with respect to each relevant variable at each line of code by applying the chain rule, and then sums up all the contributions. The FD method computes the derivatives using a Taylor series expansion, as shown in Equation C1 (Appendix C). FD computes the Jacobian matrix by looping through all the dimensions to compute the function values, perturbing with a determined step size, and computing the differences in the function. The value of the step size dictates the truncation error. Smaller step sizes reduce the error but increase the amount of numerical noise. The Complex-Step (CS) method also relies on a Taylor expansion to compute the derivatives. However, the step is represented by an imaginary term in the complex plane (Equation C3, Appendix C). The CS method typically doubles computational time, as there are two times more bits in each value. As shown in Appendix C4, we need to add a complex step to an input and run $f(x + ih\widehat{e}_j)$ to find the gradient with respect to that input. This means that all operations in f, which depends on x, need to be executed on both the real and the imaginary part. . The advantage of the CS is that the only source of error is the truncation error since there is no associated subtraction cancellation error. Adopting smaller step sizes can reduce truncation errors."

Eq. 3: The coordinates x and y in this case have the origin at the wind farm center.

Answer: Correct, and I have added text to the description of Equation (3) in line 159:

"where $k$ is an integer denoting the turbine number, $R_{wf}$ is the Wind Farm Radius, and x and y have the origin at the wind farm center."

Eq. 4-7: One axis of this parallelogram is assumed to be parallel to the x-axis.

Answer: Correct, and I have also added text to the description of Equations 4-7 in line 166:

"where $x^{UL}$, $x^{UR}$, $x^{LL}$, and $x^{LR}$ are the upper left, upper right, lower left, and lower right coordinates that respectively define the parallelogram boundaries. The upper axis of the parallelogram is assumed to be parallel to the x-axis."

Line 167: I would rearrange this paragraph to avoid confusion. A spacing constraint of 2D was adopted for the initialization of the SMAST algorithm (described in Section 3.4.1) and in the discussion of spacing constraint costs in Section 4.1.3. Then, for all other optimizations, the spacing constraint was dropped because the results show that cost does not scale well with the number of turbines.

Answer: We have attempted to follow most of the Reviewer's suggestions and re-arranged the paragraph:

"(...) A spacing constraint value of 2D (Equation 8) was adopted for the layout initialization provided by the heuristic algorithm (described in Section 3.4.1) and in the discussion of spacing constraint costs in Section 4.1.3. For the remaining optimizations, spacing constraints were disregarded and the formulation of Equation 2 was adopted. This setup for the spacing constraint is adopted because the cost of handling spacing constraints for each turbine pair does not scale well with $n_{wt}$. Additional discussion around the spacing constraint consideration in this work is provided in Section 4.1.3, where we provide a plot showing the influence of spacing constraints on speeding up GBWFLO across scales. Moreover, we show and discuss in the Discussion section that the minimum spacing in our final results is at least 1.4D."

Line 182: "The results for the number of iterations and number of multi-starts are based on more than 50,000 GBWFLOs, and these were performed with a faster setup, which uses the wind turbine, site, and wake model definitions from the IEA Wind Task 37 case study 1." Can the flow of these different methods be clarified? Some number of GBWFLOs were performed on the Horns Rev case study to understand the first objectives (time per iteration, parallelization, and constraints), and then an additional 50,000 GBWFLOs were performed on the IEA case study to address the number of iterations and number of multi-starts? I just find the wording of the sentence to be confusing.

Answer: We are going to split the general question from Reviewer 2 in sub-questions:
1) Can the flow of these different methods be clarified?

Here the authors were not 100% sure if Reviewer 2 referred to flow cases (i.e. combinations of wind speeds and wind directions) or to the 50,000 different GBWFLOs. To be safe, we here provide these two pieces of information. First of all, the flow cases (i.e. combinations of wind speeds and wind directions) are described in Table 1.

Second, there was a typo in Line 182: it should read 55,000 GBWFLOs. These cases correspond to 1,000 simulations for each combination of wind farm size ($n_{wt} = 16, 36, 64, 130, 279$) and randomness ($random_{pct} = 0, 1, 10, 50, 100$), and are shown in Figures E1 and E2 in Appendix E. Lines 308-310 in the original submission contains that information. Additionally, two extra simulations were performed for the case with $n_wt = 566$, as described in lines 300-301 to verify the linear extrapolation to larger farms. Additional 10,000 simulations with random initial layouts (not using SMAST) for each of the following wind farm sizes ($n_{wt} = 16, 64, 279$) were performed in Section 3.4.2. These were described in lines 237-240 and were used later in Section 4.3 to showcase the capabilities of SMAST.

We changed the wording in lines 182-184 and added a description of the cases in line 182 which we hope will improve the readability of the paragraph:

"The results for the number of iterations and number of multi-starts are based on more than  55,000 GBWFLOs  performed with a faster setup, which uses the wind turbine idealized 3.35MW wind turbines with constant CT, site, and wake model definitions from the IEA Wind Task 37 case study 1 (IEA Wind Task 37, 2018; Baker et al., 2019).

2) Some number of GBWFLOs were performed on the Horns Rev case study to understand the first objectives (time per iteration, parallelization, and constraints), and then an additional 50,000 GBWFLOs were performed on the IEA case study to address the number of iterations and number of multi-starts?

Answer: Correct. We described our assumptions in line 179 and again in lines 196 and 264 (time per iteration with more realistic Horns Rev I), and line 182 and again in line 298 (number of iterations and multi-starts). When we discuss total time, we again describe our assumptions in lines 348-349. We assumed a conservative approach to estimate time per iteration and assess the effect of the optimization techniques (AD, FD, CS), parallelization, and constraints. By conservative, we mean a more complete set of flow cases (23 wind speed bins and 360 wind direction bins). We used the faster IEA setup to estimate the number of iterations and multi-starts, as we needed a much larger number of simulations to showcase SMAST capabilities.

The IEA 37 setup is faster than the Horns Rev I setup because:

- windspeed vs 23 wind speeds

- As the $C_T$ is constant (ws independent) and deficit is scaled with free-stream wind speed (wake independent), we can use a non-iterative approach where the wake effects from all wind turbines to all wind turbines are calculated in one go. With the Hornsrev setup, the $C_T$ depends on the effective wind speed (free-stream wind speed minus wake effects). An iterative approach is therefore needed. In this case, we use the approach where we propagate through the wind farm in a downwind direction and calculate the wake effects from the current wind turbine to all downstream wind turbines.

Line 203: "In this work parallelization is studied, these studies are performed on a computational cluster." Sentence fragment, consider revising.

Answer: We have updated the sentence in line 203 to:

This work implemented parallelization in a computational cluster. Single node operation was utilized with each node being composed of 2x AMD EPYC 7351 16 core CPUs, @2.9GHz, with 128GB of RAM.

Line 218: What does the arrow symbol mean?

Answer: The arrow symbol was a typo in our LATEX. Please see below the updated sentence:
Next, SMAST randomly selects a point p among the points associated with the highest AEP, $\mathcal{L}_{best}$, and places the next turbine at p.

Line 242: What is the "th mean"?

Answer: This is actually a typo, thank you very much for identifying it. We have corrected lines 241-242, and they should read as:

"(...) The bandwidth in the plots represents the standard deviation within a 99% confidence interval of  the mean."

Line 249: How do you define convergence here?

Answer:
We define convergence when the AEP reaches 99.9% of the Normalized Optimized AEP (Y-axis, Figure 4). In practice, convergence would mean a flat curve in Figure 4. We have added the definition of convergence in this context by the end of the paragraph within lines 236-251:

" (...) For small problems, the random approach seems to work as the maximum AEP converges after a relatively low number of starts. Other methods are necessary for larger problems, as the example with $n_{wt} = 279$ shows that no full convergence is achieved even after 5,000 starts. Convergence when the AEP reaches 99.9% of the Normalized Optimized AEP (Y-axis, Figure 4). In practice, convergence would mean a flat curve in Figure 4."

We briefly discussed that threshold in lines 247-248 to show how many initial starts it takes for various farms to achieve 99.9% of the maximum Normalized Optimized AEP:

"Furthermore, what is noticeable in Figure 4 is that 99.9% of the maximum AEP is obtained around 500 starts for nwt = 16, around 2,500 starts for nwt = 64, and around 4,000 starts for nwt = 279. "

Figure 4: How is AEP normalized? Is it the average optimal AEP at 5,000 initial starts?

Answer: Correct, except that we consider two batches of 5,000 simulations each (10,000 simulations, as described in line 238). For each of these batches, we calculate the average optimal AEP at 5,000 initial starts. Then, we take the mean of these two values to normalize the AEP in Figure 4. We have added a brief description in lines 238 and 241:

"Aiming to showcase the capabilities of SMAST to improve multi-start for GBWFLO, we consider sets of random multi- start simulations with three different IEA 37 case studies: 16, 64, and 279 wind turbines. Those sets are the baseline for the comparisons. The methodology in this work consisted of running 10,000 random simulations (two batches of 5,000 simulations) for each case (i.e., randomly generated initial layouts), splitting the results into m chunks, and computing the maximum AEP of each chunk. Finally, the mean and confidence intervals of the m maximum values are computed. Figure 4 shows how the normalized optimized AEP varies with the number of random initial starts for the three cases. The AEP is normalized as the average optimal AEP at 5,000 initial starts at each batch. As there are two batches with 5,000 simulations for each farm size, we take the mean between these two values."

Figure 5b: The FD line is redundant because the AD and CS lines are in reference to the FD time. I also think "speedup" could be clarified in the caption as in reference to FD.

Answer: Indeed, the FD line is just a horizontal line crossing the Y-axis at 1. We deeply appreciate the comment by the reviewer. We believe that Figure 5b as it is now will not cause problems for readers to understand our message. We would like to keep the FD line, so Figure 5b follows the same standard as Figures 5a and 5c. We have added a sentence to the caption of Figure 5:

**Figure 5**. "Impact of different gradient computation methods on time per iteration, speedup, and memory usage. The speedup in Figure 5b takes FD as the basis for comparisons."

Figure 9: Can it be clarified that 'n' in the legend refers to the number of random initial starts at which SMAST results in higher optimal AEP? Also, the black dashed line is the maximum AEP of the random initial starts (i.e. the final value in Figure 4)?

Answer: First sub-question - The 'n' in the legend refers to the number of initial starts at which SMAST results in higher Normalized Optimized AEP as compared to 10,000 simulations that used an approach with layouts randomly initialized. We have added the description to the legend of Figure 9:

**Figure 9**. "Normalized Optimized AEP as a function of the number of initial starts, considering the

IEA 37 study-case. The 'n' variable refers to the number of initial starts at which the SMAST approach results in higher Normalized Optimized AEP as compared to 10,000 simulations that used an approach with layouts randomly initialized."

The second sub-question - Correct, the black dashed in Figure 9 corresponds to the maximum normalized AEP from Figure 4, which is based on the maximum AEP at 5,000 initial starts with randomly initialized layouts. As we ran two sets of these 5,000 simulations (10,000 total), we took the mean of the two maximum AEP values from each batch. These are the values in the black dashed lines in Figure 9. Lines 328-330 provide an explanation of the black dashed lines:

"The black dashed lines in each plot show the maximum AEP, and the gray bands a 99% confidence interval of the best result for the optimized AEP among two sets of 5,000 simulations with entirely 330 random initial layouts ($random_{pct} = 100$). "

Line 349: "more realistic [Horns Rev I] setup"

Answer: We have changed line 349 as below:

"the $t_{total}$ is estimated using Equation 1 and combines the iteration time found with the more realistic  setup (Hornsrev1) with the number of iterations and multi-starts found with the faster IEA37 setup."

Line 350: "Assuming that the result on number of multi-starts for 279 wind turbines is similar for a wind farm with 500 wind turbines." Sentence fragment, consider revising.

Answer: We have revised the sentence in Line 350 as below:

" Additionally, we assume that the result on a number of multi-starts for 279 wind turbines is similar for a wind farm with 500 wind turbines."

Line 389: What are "wriggles"?

Answer: We use the word "wriggle" to describe direction-dependent variations in the wind turbine wakes when averaging all the wind directions with their sector frequency weight. We have included this definition in the paragraph within lines 387-392:

"Another possibility to speed up wind farm optimization is considering a subset of the flow cases. For wind farm AEP computations, Thomas et al. (2022b) demonstrated that at least 40 or 50 wind sectors are necessary to run WFLO. However, wriggles can occur when simulating too few wind directions. We define wriggles as direction-dependent variations in wind turbine wake when averaging all the wind directions with their sector frequency weight. In simple cases, the occurrence of such wriggles has been found to drastically increase the number of local minima. Taking this into consideration, in this work, we took the conservative approach of considering wind direction bins of 1°. However, we acknowledge other possibilities and intend to explore them in future work."